# Personalized Image Generation via Human-in-the-loop Bayesian Optimization

**Rajalaxmi Rajagopalan** [1]  **Debottam Dutta** [1]  **Yu-Lin Wei** [1]  **Romit Roy Choudhury** [1]

## Abstract

Imagine Alice has a specific image $x^*$ in her mind, say, the view of the street in which she grew up during her childhood. To generate that exact image, she guides a generative model with multiple rounds of prompting and arrives at an image $x^{p*}$. Although $x^{p*}$ is reasonably close to $x^*$, Alice finds it difficult to close that gap using language prompts. This paper aims to narrow this gap by observing that even after language has reached its limits, humans can still tell when a new image $x^+$ is closer to $x^*$ than $x^{p*}$. Leveraging this observation, we develop **MultiBO** (Multi-Choice Preferential Bayesian Optimization) that carefully generates $K$ new images as a function of $x^{p*}$, gets preferential feedback from the user, uses the feedback to guide the diffusion model, and ultimately generates a new set of $K$ images. We show that within $B$ rounds of user feedback, it is possible to arrive much closer to $x^*$, even though the generative model has no information about $x^*$. Qualitative scores from 30 users, combined with quantitative metrics compared across 5 baselines, show promising results, suggesting that multi-choice feedback from humans can be effectively harnessed for personalized image generation.

## 1. Introduction

Modern diffusion models like FLUX (Labs et al., 2025), StableDiffusion3 (Esser et al., 2024), Lumiere (Bar-Tal et al., 2024) continue to make remarkable advances in image generation. In tandem, users continue to raise the bar, asking for images to not only be of high quality, but to also accurately obey their language prompts; the expectation is that the generated image will match an image $x^*$ that the user has in her mind. Can generative models deliver on this expectation? Perhaps detailed surgery on generative model outputs, using a careful combination of prompting, masking,

editing, inpainting, etc. could produce the desired image. However, for most lay users, language based prompting is the main form of expression and that may be inadequate for hyper-personalization. This is expected because the space of images is much higher dimensional than language, hence a given prompt maps to many possible images. Moreover, humans have limited vocabulary and expressivity in describing an image, so not everyone will be able to perfectly craft the optimal prompt. Finally, in the near term, models that jointly learn language and image representations (e.g., CLIP) are likely to have imperfections, causing further misalignment between language and images. Assuming these are true, it appears that there will be a fundamental gap between the image $x^*$ the user has in mind, and the best image $x^{p*}$ that the user can generate through language prompts $p$. Can we narrow down this gap for everyday users?

Observe that even after language-based prompting has saturated, if a diffusion model could present some images better than $x^{p*}$, humans can quite robustly pick the image closest to their target $x^*$. This preference indication carries valuable information to narrow the gap. We cast this as a human-in-the-loop black box optimization problem with preferential input. The goal is to achieve free-form, training-free and interactive image personalization, specifically when language prompting has neared saturation.

We are not the first to utilize human preference in diffusion models; a rich body of work in the areas of *reward-based alignment*, *preference alignment*, and *reinforcement learning using human feedback (RLHF)* is actively exploring this space of ideas (Xu et al., 2023; Wallace et al., 2024; Yeh et al., 2024; Song et al., 2021). Section 5 discusses them in detail with the closest being DEMON (Yeh et al., 2024), a training-free inference-time approach that optimizes the noise at each $t$ without back propagation, using a stochastic optimization by leveraging Probability Flow ODE (PF-ODE) (Song et al., 2021). Our main departure from past work lies in upgrading user preference to a multi-choice format, allowing the user to select any subset from $K$ image options. This offers much richer information, but absorbing this information into blackbox optimization requires a redesign of the likelihood model and the acquisition function. We conduct this redesign and demonstrate that **MultiBO**'s image generation gets tailored to each individual, is not encumbered by reward-hacking, and does not

[1]University of Illinois, Urbana-Champaign. Correspondence to: Rajalaxmi Rajagopalan <rr30@illinois.edu>.

*Proceedings of the $43^{rd}$ International Conference on Machine Learning*, Seoul, South Korea. PMLR 306, 2026. Copyright 2026 by the author(s).

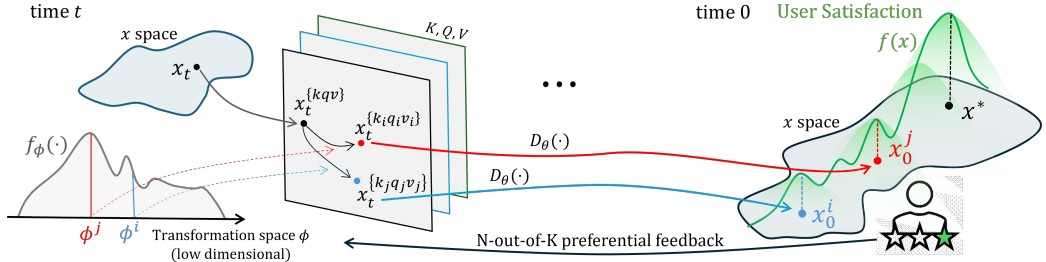

*Figure 1.* The flow of ideas in **MultiBO**: The BO optimization presents the user with $K$ images and the user chooses $N$ out of $K$ images based on closeness to user's imagined image $x^*$. BO accepts the $N$-out-of-$K$ preferential user feedback and optimizes on the space of transformations applied to the self-attention $KQV$ features, to generate next round of $K$ images, iteratively moving closer to $x^*$.

need offline training using large datasets. Our users have to indeed wait between each round of feedback, but assuming that is tolerable, the generated image better approaches the user's imagined image, $x^*$. We sketch our core ideas below.

In standard diffusion, $D_\theta(.)$ denoises a noisy image $x_t$ to $x_0$. Our goal is to arrive to $x^*$ by performing a human-in-the-loop optimization on $x_t$ as follows:

$$\arg\min_v f(x^*) - f(D_\theta(x_t + v)) \tag{1}$$

Here $f$ is the user's satisfaction function, maximized at $x^*$. Unfortunately, $f(.)$ is unknown—a blackbox function— hence there is no gradient to be obtained. However, it is possible to sample $f(.)$, meaning that we can query the user for feedback for any given image $D_\theta(x)$. Using such feedback, we can adopt existing Bayesian optimization (BO) methods using Gaussian Process Regression (GPR), explained in Sec. 2. Briefly, BO will propose judiciously chosen variants of $x_t$, say $x_t^j$, and the user will rate how close $D_\theta(x_t^j)$ is to $x^*$. BO can utilize this feedback to update its predictions, eventually finding $\hat{x}_t^*$ such that $D_\theta(\hat{x}_t^*)$ is arbitrarily close to $x^*$. Of course, to reduce user burden, the number of user queries needs to be limited to a budget $B$.

Problems arise in realizing this high level idea. ❶ Operating BO in the high dimensional pixel space is very difficult. The optimization must be re-cast to a much lower dimensional space, while ensuring that the manifold of correct images, at time $t$, can be reached from $x_t$. ❷ Giving a numerical feedback, $f(D_\theta(x_t^j))$, is known to be difficult for users (Tsukida & Gupta, 2011) because they may not be able to *quantify* how much worse a given $D_\theta(x_t^j)$ is compared to $x^*$. However, it is easier to express preferences between pairs or sets of $K$ images. A user-friendly solution must design the BO framework in a suitable low-dimensional space, incorporating the user's preferential feedback.

Past work have made progress where diffusion models leverage human preference. For example, (Xu et al., 2023; Fan et al., 2023; Wallace et al., 2024) train (global) reward models, where offline volunteers express preferences between pairs of images; this reward model becomes a proxy for

human preference, which ultimately guides the denoising vector. Similar ideas exist in blackbox optimization through preferential likelihood models (Chu & Ghahramani, 2005), where GPR accommodates pairwise preferential feedback from users (more in Sec. 2). However, when the function domain is high dimensional, and when a user's preference is pairwise, mapping the function reasonably well incurs excessive user feedback. Reducing the user burden calls for richer preferential information, and then integrating them into a lower dimensional space for optimization speed up.

***Key Contributions:*** **MultiBO** contributes by observing that *multi-choice preference queries* bring far richer information to the optimization, and shows that such N-out-of-K feedback can be mathematically accommodated by GPR through an updated likelihood model and acquisition function. Moreover, building on the empirical success of (Nam et al., 2024), **MultiBO** proposes to perform the GPR optimization in a low-dimensional transformation space, where the transformations are applied on $\langle K, Q, V \rangle$ matrices in the attention layer of a diffusion model. The family of transformations are parameterized by $\phi$ in suitably low dimensions (see Figure. 1), but offer flexibility to explore the image manifold around $x_t$. In sum, **MultiBO** utilizes multi-choice user feedback to optimize image representations inside the attention layer, empowering users to generate images close to their imagination.

Experiments are designed with a human picking a target image $x^*$, and a starting image $x^{p*}$ that bears similarity to $x^*$ (i.e., language prompts cannot easily convert $x^{p*}$ to $x^*$). Five baselines attempt to optimize towards $x^*$, either with real human feedback, or with their respective (pre-trained) reward models. At the end, each baseline submits their final image $\hat{x}^*$ and 30 external volunteers evaluate the closest match (and other opinion scores). We also use quantitative metrics and a variety of ablations to shed light on the internal pros and cons of each method, and our own design choices. Results show that **MultiBO** robustly outperforms other methods and there is room for further improvement.

## 2. Background

■ **Bayesian Optimization**: Bayesian optimization (BO) (Frazier, 2018; Wang, 2020) is a non-parametric black-box optimization method that consists of two modules:

(1) *Gaussian Process Regression (GPR)*: constructs a probabilistic surrogate of $f$ using the Gaussian likelihood model. The GPR posterior captures our beliefs about the unknown objective function. Given a set of observations $(\mathcal{X}, \mathcal{F})$ and the Gaussian prior kernel, $\mathbf{K}$, the GPR posterior is,

$$P(\mathcal{F}|\mathcal{X}) \sim \mathcal{N}(\mathcal{F}|\boldsymbol{\mu}, \mathbf{K}) \quad (2)$$

where, $\boldsymbol{\mu}$ is the best model of the function $f$ and $\mathbf{K}$ represents the uncertainty map of $f$ over unsampled regions of the input space $\mathcal{Y}$.

(2) *Acquisition function (ACF)*: a sampling strategy that seeks to identify future observations that would improve the likelihood model. We use "Expected Improvement" (EI) ACF:

$$x_{\mathbf{EI}} = \arg\max_{x \in \mathbf{R}^D} \mathbf{E}[[f(x) - f^*]^+ | \mathcal{X}, \mathcal{F}] \quad (3)$$

where, $\mathbf{E}[\cdot|\mathcal{X}, \mathcal{F}]$ is the expectation on the GPR posterior (Eqn. 2) given observations $(\mathcal{X}, \mathcal{F})$.

■ **Preferential Bayesian Optimization**: Let us first consider the case of paired preference data i.e., the user is presented two choices to pick one from. A user's preference information (favoring A over B) can be used to construct a likelihood model. The *probit* likelihood model (Thurstone; Mosteller, 1951) allows us to infer $f$ from the binary preference observations. Assume we have shown the user $M$ pairs of data from a set of $N$ samples $[x_1, \ldots, x_N]$. The data set therefore consists of the ranked pairs:

$$\mathcal{X} = \{a_i \succ b_i; i = 1, \ldots, M\} \quad a_i, b_i \in \mathcal{Y} \quad (4)$$

Assuming noisy observations, the latent function values are, $v(a_i) = f(a_i) + \delta, \quad v(b_i) = f(b_i) + \delta, \delta \sim \mathcal{N}(0, \sigma^2)$ is Gaussian noise. Following the GPR modeling of $f$ (Eqn. 2), The Bradley-Teller-Luce (BTL) (Stein, 1999) model defines the probability that data $a$ is preferred to data $b$ as,

$$P(a_i > b_i | f(a_i), f(b_i))$$
$$= \int \int P(a_i > b_i | f(a_i) + \delta_a, f(b_i) + \delta_b) \quad (5)$$
$$\cdot \mathcal{N}(\delta_a; 0, \sigma^2)\mathcal{N}(\delta_b; 0, \sigma^2)d\delta_a d\delta_b = \Phi(z_i)$$

where $z_i = \frac{f(a_i) - f(b_i)}{\sqrt{2}\sigma}$ and $\Phi(z) = \int_{-\infty}^{z} \mathcal{N}(\gamma; 0, 1)d\gamma$ is the Gaussian CDF. This preferential likelihood model is called a **Binomial-Probit** regression model. It can be converted to a **Binomial-Logit** model by setting $\varphi(z_i) = \frac{1}{1+\exp(-z_i)}$. Given $M$ pairwise observations, The BTL logit

likelihood model of observing the preference relations given the latent function values $f(x_i)$ is,

$$P(\mathcal{X}|\mathbf{f}) = \prod_{i=1}^{M} P(a_i > b_i | f(a_i), f(b_i)) \quad (6)$$

and the GPR posterior is,

$$P(\mathbf{f}|\mathcal{X}) = \frac{P(\mathbf{f})}{P(\mathcal{X})} \prod_{i=1}^{M} P(a_i > b_i | f(a_i), f(b_i)) \quad (7)$$

where $P(\mathbf{f})$ is the prior, and $P(\mathcal{X}) = \int P(\mathcal{X}|\mathbf{f})P(\mathbf{f})d\mathbf{f}$. The detailed proofs are well established in (Chu & Ghahramani, 2005) and more details are in Appendix A.1.1.

This likelihood serves as the pathway for probabilistic modeling of human preference and its optimization using the Bayesian framework. Thus, BO applied to user preference data is Preferential BO (PBO) (Chu & Ghahramani, 2005). Additional background on diffusion and attention are in Appendix A.2.

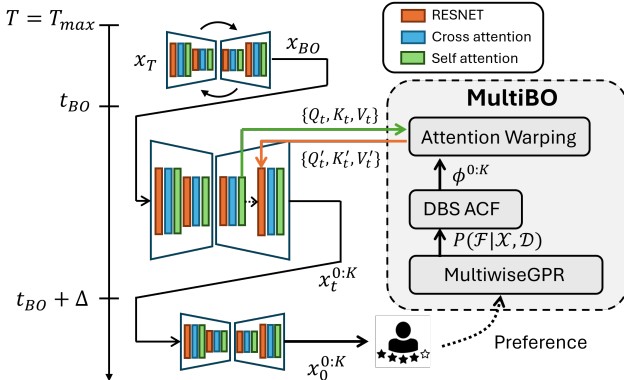

*Figure 2.* **MultiBO** Image Personalization Pipeline. **MultiBO** optimizes the self attention $Q, K, V$ features at time interval $[t_{\text{BO}}, t_{\text{BO}} + \Delta]$ over warping transformation space $\mathcal{Y}$. At each iteration $i$, **MultiBO** offers $K_i$ transform parameter choices and the user picks the $N$ "best" option(s) from the corresponding $K$ attention-modified images. **The MultiwiseGPR** likelihood models the unobservable user *satisfaction* function $f$ from the user preferences and the **Dynamic Balanced Subspace (DBS)** acquisition function prescribes the next set of $K_{i+1}$ warp parameters.

## 3. MultiBO

Briefly, we aim to adjust diffusion generation $x_0 = D_\theta(x_t)$ to closely match a target $x^*$. We cast this as a human-in-the-loop Black-box optimization (BBO) problem and use Bayesian Optimization (BO) approaches. As discussed in Sec. 2, Preferential Bayesian Optimization (PBO) provides the framework for taking in user feedback in the form of paired preference and modeling unknown *user satisfaction* function $f$ using Pairwise GPR (Eqn. 5). Optimizing the corresponding GPR posterior (Eqn. 7) produces $\hat{x}^*$.

Two challenges arise in our problem: (1) we can only obtain finite user feedback (typically 50 questions). Therefore,

each user preference feedback has to reveal as much information as possible about unknown $f$. (2) BO has to operate on very high-dimensional RGB Pixel Space to find $\hat{x}^*$, unsuitable for BBO. Our algorithm **MultiBO** addresses these via two proposed modules: *Multi-choice Preferential Bayesian Optimization* and *Self-Attention Warping*, as illustrated in Figure 2.

### 3.1. Multi-choice Preferential BO (Multi-choicePBO)

In PBO, for the PairwiseGPR model to be a good approximation for $f$, many observations (preference feedback) are required. An observed pair only provides information about $f$ (resolves uncertainty) for two halves of the input space $\mathcal{Y}$ and more data is needed to sufficiently map the function landscape. This is further exacerbated by increasing dimensionality of the input space. Large number of queries will put considerable burden on the user, diminishing feasibility for human-in-the-loop setting.

To reduce this user burden is to increase the information extracted from each user preference. One way to achieve this is to expand the choice set to $K$ images. This makes intuitive sense, as each observation now resolves $f$ on $K$ partitions of the input space $\mathcal{Y}$. The user has the flexibility to choose $N$-out-of-$K$ best images, $N > 1$ if the user thinks they are equally good or they contain complementary aspects related to the target image. This can significantly bring down the number of user queries at the expense of slight increase in user's cognitive load of selecting $N$-out-of-$K$ images instead of 1-out-of-2.

To handle this complex preference signal, we design *MultiwiseGPR*, a likelihood model that maps multi-choice preference relations to $f$ and *Dynamic Balanced Subspace (DBS)* acquisition function that supplies the next set of $K$ choices based on the MultiwiseGPR posterior.

■ **MultiwiseGPR**: We take the discussion on the more general $N$-out-of-$K$ setting to the Appendix A.1.3. Consider the special case when $N = 1$ i.e., user picks 1 image out of $K$ choices. Consider a set of $N$ distinct samples $x_i \in \mathcal{Y} \subseteq \mathbb{R}^D; [x_i : i = 1, ..., N]$. Let $\mathcal{Z}$ denote the $K$ choices chosen from it for one observation set. The set of $M$ observed multiwise preference relations on the choice set $\mathcal{Z}_i, i = 1, \ldots, M$ is,

$$\mathcal{X}_i = \{a_i^{(w)} \succ \{a_i^{(j)}\}_{j \in \mathcal{Z}_i/w} : i = 1, ..., M\} \quad (8)$$

where $\{a_i\}_{\mathcal{Z}_i} \in [x_1, \ldots, x_N]$ and $a_i^{(w)} \succ \{a_i^{(j)}\}_{j \in \mathcal{Z}_i/w}$ means $a_i^{(w)}$ is the preferred winning sample over $K - 1$ other choices in choice set $\mathcal{Z}_i$. This preference relationship is modeled by a polycotomous regression model (Held & Holmes, 2006) – multinomial-logit regression model.

*Multinomial-Logit Regression*: Let us consider the simple case when $M = 1$ observation. There are $K$ samples (images) in the choice set $\{a_1, \ldots, a_K\} \in \mathcal{Z}$. We use a

Gaussian prior, and the corresponding latent function values are $\mathbf{f} = (f(a_1), \ldots, f(a_K))$.

If we assume IID Gaussian noise $\mathcal{N}(0, \sigma)$, then, $v_j = f_j + \delta_j, \delta \in \mathcal{N}(0, \sigma)$. Picking choice $a_i^{(w)}$ out of $\{a_i^{(j)}\}_{j \in \mathcal{Z}_i}, i = 1, \ldots, M$ is modeled by a categorical distribution – multinomial-probit (multi-choice version of the binomial in equation 5) as,

$$P(Y = w|\mathbf{f}) = P(v_w = max_j v_j)$$
$$= \int \mathbf{1}\{f_w + \delta_w \geq f_j + \delta_j, \forall j \neq w\}\Phi(\delta)d\delta \quad (9)$$

where, $Y$ is picking winning index $w$ in $K$ choices $\mathcal{Z}$. This is a multivariate normal orthant probability (multidimensional CDF). There is no closed-form expression for this multinomial-probit regression model. Following PBO (Chu & Ghahramani, 2005), we replace the probit model for the logistic model.

Let's update our assumption to IID $\delta \in \text{Gumbel}(0, 1)$ noise in the observation, the multinomial-logit distribution is,

$$P(Y = w|\mathbf{f}) = \exp(f_w)/ \sum_{j=1}^{K} \exp(f_j) \quad (10)$$

The logistic distribution is characterized by the *softmax* function (parallels sigmoidal definition in Eqn. 5). The joint likelihood of observing $M$ multi-choice observations given the latent function values $f(x_i)$ is the product of the likelihood function of each observation in Eqn. 10,

$$P(\mathcal{X}|\mathbf{f}) = \prod_{i=1}^{M} P(Y = w|\mathbf{f}) \quad (11)$$

and the corresponding Multiwise GPR posterior is,

$$P(\mathbf{f}|\mathcal{X}) = \frac{P(\mathbf{f})}{P(\mathcal{X})} \prod_{i=1}^{M} P(Y = w|\mathbf{f}) \quad (12)$$

where $P(\mathbf{f})$ is the Gaussian prior, and $P(\mathcal{X}) = \int P(\mathcal{X}|\mathbf{f})P(\mathbf{f})d\mathbf{f}$. The proofs for the likelihood and estimation of the posterior are found in Appendix A.1.2 and (Held & Holmes, 2006). Thus, we have a probabilistic mapping between multi-choice preference and $f$ that is leveraged by the acquisition function in finding $\hat{x}^* = \arg \max f$.

■ **Dynamic Balanced Subspace (DBS)**: MultiwiseGPR expects future observations $\mathcal{X}$ that improve the belief of $f$ (Eqn. 12). The naive way is to extend the Expected Improvement (EI) ACF (Eqn. 3) to $K$-EI i.e., $K$-jointly maximize the posterior expectation. This joint optimization of an already intractable expectation incurs a tremendous computational load (especially when $K$ is large). In human-in-the-loop settings, we design a light ACF offering $K$ "good" choices that balance observed preferences (*exploitation*) with diverse alternatives (*exploration*).

DBS ACF bypasses the $K$-EI computational trap by computing only 1 sample, $x_{\text{EI}}$. Armed with $x_{\text{EI}}$ and the the current

best sample $\hat{x}^*$, DBS ACF constructs a set of $K$ anchor points that act as *bridge vectors* $\mathbf{v}_{bridge}$ connecting them,

$$\mathbf{v}_{bridge,i} = \hat{x}^* + \gamma_i(x_{\text{EI}} - \hat{x}^*), \gamma_i \in [0, 1] \quad (13)$$

The key intuition is that $\mathbf{v}_{bridge}$ modulate the explore-exploit trade-off between BO's forward thrust $x_{\text{EI}}$ and the belief $\hat{x}^*$ ensuring that the preference set $\mathcal{Z}$ always includes candidates that represent a direct transition from the best known $\hat{x}^*$ to the theoretical optimal global point $x_{\text{EI}}$. The final $K$ choices $(x^+)$ presented to the user are constructed by random perturbations $\delta$ along the $\mathbf{v}_{bridge}$ directions,

$$x_i^+ = \text{proj}_{\mathcal{Y}}\left(\mathbf{v}_{bridge,i} + \delta_i\right), \quad i = 0, \ldots, K - 1 \quad (14)$$

where $\text{proj}_{\mathcal{Y}}$ is the projection onto the input space $\mathcal{Y} \subseteq \mathbb{R}^D$. As we are cognizant of the user burden constraints on the optimization especially when the input space $\mathcal{Y} \subseteq \mathbb{R}^D$ is high-dimensional, DBS ACF samples the perturbations $\delta \in \Omega$ from a subspace $\Omega \subseteq \mathcal{Y}$, instead of $\mathcal{Y}$. Borrowing from the work Bounce (Papenmeier et al., 2023) that performs high-dimensional Bayesian Optimization using embedding spaces of increasing dimensionality, we design our DBS ACF to operate over a subspace constructed dynamically as the optimization proceeds.

***d-dimensional Subspace***: We identify the most influential directions in the $D$-dimensional input space $\mathcal{Y}$ by constructing an uncentered covariance matrix $\mathbb{C}$ of the GPR posterior gradients. Let $\mu(\mathbf{x})$ denote the posterior mean of the Multiwise GPR model. We estimate $\mathbb{C}$ by averaging the outer products of the gradients at $H$ design points sampled in the neighborhood of the current best point $\hat{x}^*$ as,

$$C = \frac{1}{H} \sum_{i=1}^{H} \nabla\mu(\mathbf{x}_i)\nabla\mu(\mathbf{x}_i)^T, x_i = \hat{x}^* + \eta \quad (15)$$

We determine the subspace dimension $d$ by the **Spectral Gap**, following the eigendecomposition of $\mathbb{C}$,

$$d = \arg\max_i \lambda_i/\lambda_{i+1} \quad i \in \{1, 2, \ldots, D\} \quad (16)$$

This identifies the threshold $d$ after which additional dimensions contribute significantly less to the visual change of the image, allowing DBS ACF to adaptively *unlock* more dimensions when the function $f$ landscape is complex and *collapse* to a lower-dimensional space when a few dimensions dominate the visual features. Hence, the $d$-dim. subspace $\Omega$ represents a $d-$volume in the $D$-dim. input space that contains the most visually significant dimensions. The perturbations $\delta$ in Eqn. 14 are now samples from the $d-$Subspace $\Omega$,

$$\delta_i = \sigma \sum_{j=1}^{d} \alpha_{i,j}\sqrt{\lambda_j}\mathbf{u}_j \quad (17)$$

where, each coordinate $\alpha_{i,j} \in [-1, 1]$ is *Eigen-Weighted* by scaling with corresponding eigenvalue $\lambda_j$. This ensures that perturbations along the most sensitive dimensions are more pronounced than those along dimensions with lower

influence. The eigen weighting and perturbations along the bridge vector ensure that the $K$ choice set of $\mathcal{Z}$ does not collapse into a redundant line between the anchors $\{\hat{x}^*, x_{\text{EI}}\}$, i.e., even if $\mathbf{x}_{\text{EI}}$ and $\hat{x}^*$ are close in the input space, the user is presented with diverse visual variations spanning the most influential latent dimensions of the diffusion model space.

## 3.2. Self-Attention Warping

Theoretically, Multi-choicePBO would operate on diffusion's latent space $x_t \in \mathbb{R}^D$. However, this is practically infeasible as the latent space dimensions $D$ are $\approx 16k$. Attention mechanism in diffusion influences both local and global semantic and structural attributes of a generated image. Thus, Multi-choicePBO optimization on the attention features offers the unique opportunity to affect both global and local changes without the dimensionality cost. As the user has already constructed the most expressive prompt, cross-attention features are not relevant to the task at hand. Instead, we optimize on self-attention $Q, K, V$ features that controls spatial features of the image as well as attributes like texture, shape, color, etc.

Keeping the practical constraints of human-in-the-loop optimization in mind, we are interested in further constraining the Multi-choicePBO optimization in $Q, K, V$ space. A reasonable constraint design restricts optimization to valid transformations of the attention space. Since the attention mechanism directly correlates with the pixel-level features of the image, we employ a family of transformations typically applied to images – Warping (Truong et al., 2021).

A ***Warping*** transform is a functional mapping of each pixel from a reference image to a transformed image. This parallels **MultiBO**'s optimization goal of finding that transformation that maps $x_0$ to the target $x^*$. Affine transformations are linear and provide global alignment; while Thin Plate Spline transforms (TPS) are non-linear deformations that enable local refinement.

*Affine*: $\mathbf{x}' = A\mathbf{x} + \tau$ where $A \in \mathbb{R}^{2 \times 2}$ and $\tau \in \mathbb{R}^2$.
*Thin-Plate Spline (TPS) Transformation*: $f(\mathbf{x}) = a_1 + a_x x + a_y y + \sum_{i=1}^{N} w_i U(\|\mathbf{x} - \mathbf{c}_i\|)$, with $U(r) = r^2 \ln(r^2)$.

**MultiBO** warps self-attention $Q, K, V$ features using a composed transform of affine and TPS, *Affine + TPS Composition*: $I_q^{\text{warp1}} = \text{Affine}(I_q), \quad I_q^{\text{warp2}} = \text{TPS}(I_q^{\text{warp1}})$.

Thus, **MultiBO**'s attention optimization problem is,

$$Qs_t^*, Ks_t^*, Vs_t^* = \arg\max_{\Theta \in \mathcal{Y}} \quad \text{W}(Qs_t, Ks_t, Vs_t; \Theta) \quad (18)$$

where, $\mathbf{W}(\cdot)$ is the warping function, the constrained optimization search space, $\mathcal{Y}$ is the Affine+TPS warping space rather than $\mathbb{R}^{H \times W \times d}$ and $t$ is the diffusion timestep. The optimization framework is Multi-choicePBO. The image personalization pipeline of **MultiBO** is presented in Algorithms 1, and 2 in Appendix A.3.

$x_0$ $\quad$ $\hat{x}^*_{B=10}$ $\quad$ $\hat{x}^*_{B=20}$ $\quad$ $\hat{x}^*_{B=30}$ $\quad$ $\hat{x}^*_{B=50}$ $\quad$ $x^*$

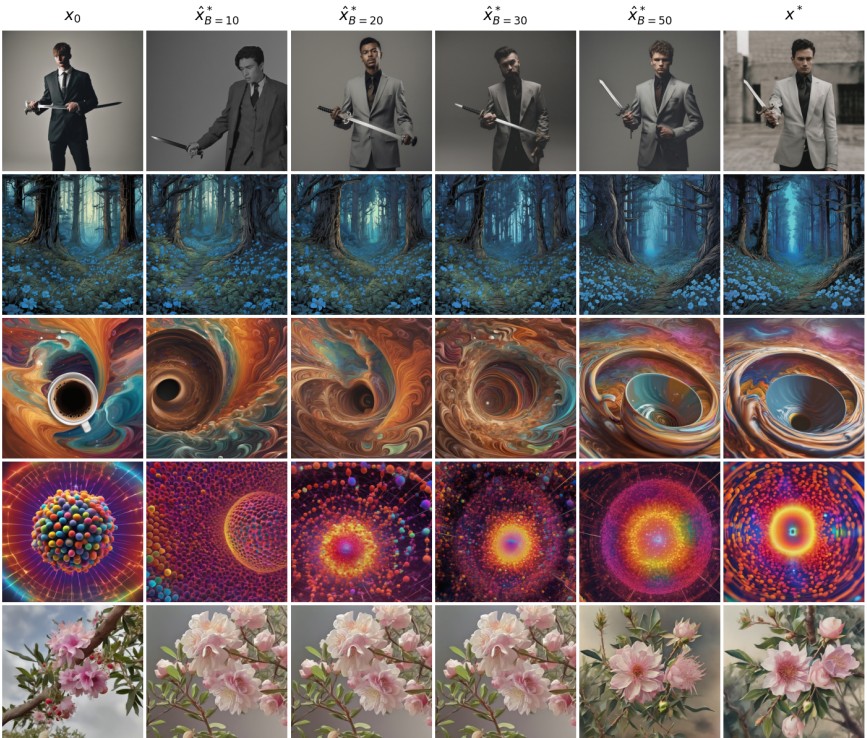

*Figure 3.* Qualitative Results– **MultiBO** optimization progress: Starting image $x_0 = D_\theta(x_t)$ after prompting, best image $\hat{x}^*$ after $B = 10, 20, 30, 50$ iters, and the true target $x^*$. (Due to space constraints, the corresponding prompts are in the Appendix A.4.2).

# 4. Experiments

■ **Datasets**: We curate prompts and corresponding target images $x^*$ from popular prompt datasets in diffusion image editing space like T2ICompBench (Huang et al., 2023b) and preference alignment space such as HPSv2 (Wu et al., 2023) benchmark dataset, PartiPrompts (Yu et al., 2022), etc.

■ **Baselines**: (1) *Preference Alignment*: We compare with training-based methods: DiffusionDPO (Wallace et al., 2024) and IterComp (Zhang et al., 2024a), training-free methods: DNO (Tang et al., 2024), DAS (Kim et al., 2025), and DEMON (Yeh et al., 2024).
(2) ***MultiBO***$_{<reward>}$: Replace the human scorer with popular reward scores and target-based metrics, CLIP-I2I and LPIPS. This category of baselines is closest to the alignment works except with a different optimization (BO) algorithm.
(3) *L2-guided*: the ideal upper limit, constructed by directly guiding diffusion denoising with classifier guidance in the form of L2 loss w.r.t target $x^*$.
(4) ***MultiBO***$_{Subspace}$: We replace attention space optimization with **MultiBO** applied on a $100D$ subspace of $x_t$ (Following (Chen et al., 2024)).

■ **Metrics**: We compare **MultiBO** against popular preference alignment reward metrics like PicScore (Kirstain et al., 2023), HPSv2 (Wu et al., 2023), Aesthetic (Schuhmann & Beaumont, 2024), ImageReward (Xu et al., 2023), and VILA (Ke et al., 2023). For target alignment, we use

CLIP I2I, and LPIPS w.r.t $x^*$. The implementation details of **MultiBO** along with datasets and baselines are provided in Appendix A.4 and here[1].

## 4.1. Results

The two key performance goals of **MultiBO**: Alignment with (1) target $x^*$, (2) general human preference metrics.
❶ **Alignment with Target** $x^*$: Figure 3 qualitatively tracks **MultiBO**'s optimization progress. After $B = 50$ iterations, **MultiBO** steers $D_\theta$ to produce $\hat{x}^*$ well aligned with $x^*$. **MultiBO** exhibits superior performance in target-based metrics, CLIP-I2I and LPIPS in Table 1 over all baselines except L2-guided, which is the upper-bound on how close we can get to the target (except DEMON$_{LPIPS}$). Preference Alignment baselines were not tuned for a particular target so their poor CLIP-I2I and LPIPS performance is expected. However, the more interesting result is that **MultiBO** also beats **MultiBO**$_{CLIP-I2I}$ and **MultiBO**$_{LPIPS}$ (**MultiBO**'s human scorer replaced by the corresponding metric scorer and directly taking target as input). This validates our hypothesis that humans function as sophisticated guidance models, capturing nuances missed by approximate metrics. Notably, users dynamically shift focus between global semantics and precise local spatial control through their preference choices.

---
[1] https://annonanom125.github.io/ annonAnomrepo/index.html

*Table 1.* Quantitative comparison of different methods on the personalized image generation task (**Top**, second best, third best).

| | Properties | | Target Alignment | | Reward Metrics | | | | Image Quality |
|---|---|---|---|---|---|---|---|---|---|
| | training-free | model-agnostic | CLIP-I (↑) | LPIPS (↓) | AES (↑) | Picscore (↑) | HPSv2 (↑) | ImageReward (↑) | VILA (↑) |
| L2-guided (Oracle) | ✓ | ✓ | 0.9811 | 0.1958 | 6.2163 | 0.2201 | 0.2646 | 0.5883 | 0.6158 |
| IterComp | ✗ | ✗ | 0.8539 | 0.6988 | 6.1879 | 0.2289 | 0.2712 | 1.1835 | 0.6708 |
| DiffusionDPO | ✗ | ✗ | 0.8365 | 0.7535 | 6.2367 | 0.2245 | 0.2640 | 0.6029 | 0.6560 |
| DEMON$_{Aesthetic}$ | ✓ | ✓ | 0.8109 | 0.6785 | 7.2685 | 0.2224 | 0.2644 | 0.4902 | 0.6484 |
| DNO$_{Aesthetic}$ | ✓ | ✓ | 0.6931 | 0.8487 | 7.7674 | 0.1905 | 0.2517 | -0.9815 | 0.4509 |
| DAS$_{Aesthetic}$ | ✓ | ✓ | 0.6849 | 0.8495 | 7.7032 | 0.1899 | 0.2498 | -1.2349 | 0.4514 |
| **MultiBO$_{Aesthetic}$** | ✓ | ✓ | 0.8839 | 0.6418 | 7.2313 | 0.2212 | 0.2634 | 0.7299 | 0.6649 |
| DEMON$_{PicScore}$ | ✓ | ✓ | 0.8839 | 0.6742 | 6.7685 | 0.2221 | 0.2637 | 0.4805 | 0.6792 |
| DNO$_{PicScore}$ | ✓ | ✓ | 0.8828 | 0.6449 | 6.2013 | 0.2326 | 0.2731 | 0.9812 | 0.6460 |
| DAS$_{PicScore}$ | ✓ | ✓ | 0.8758 | 0.6633 | 6.2190 | 0.2321 | 0.2724 | 0.9553 | 0.6425 |
| **MultiBO$_{PicScore}$** | ✓ | ✓ | 0.8877 | 0.6411 | 6.2282 | 0.2286 | 0.2679 | 0.8856 | 0.6752 |
| DEMON$_{HPSv2}$ | ✓ | ✓ | 0.8472 | 0.6795 | 7.2777 | 0.2218 | 0.2627 | 0.4922 | 0.6721 |
| DAS$_{HPSv2}$ | ✓ | ✓ | 0.8298 | 0.7436 | 5.9925 | 0.2128 | 0.2673 | 0.6852 | 0.6070 |
| **MultiBO$_{HPSv2}$** | ✓ | ✓ | 0.8818 | 0.6474 | 7.1735 | 0.2230 | 0.2722 | 0.8695 | 0.6806 |
| DEMON$_{ImageReward}$ | ✓ | ✓ | 0.8520 | 0.6824 | 6.7221 | 0.2232 | 0.2641 | 1.1530 | 0.6642 |
| **MultiBO$_{ImageReward}$** | ✓ | ✓ | 0.8794 | 0.6468 | 6.7310 | 0.2221 | 0.2664 | 1.2275 | 0.6655 |
| DEMON$_{LPIPS}$ | ✓ | ✓ | 0.9095 | 0.5907 | 6.2229 | 0.2184 | 0.2623 | 0.3765 | 0.6154 |
| **MultiBO$_{CLIP-I2I}$** | ✓ | ✓ | 0.9246 | 0.6479 | 6.0338 | 0.2198 | 0.2639 | 0.8206 | 0.6186 |
| **MultiBO$_{LPIPS}$** | ✓ | ✓ | 0.9114 | 0.5924 | 6.1711 | 0.2197 | 0.2623 | 0.8414 | 0.6250 |
| **MultiBO$_{Subspace}$** | ✓ | ✓ | 0.8014 | 0.6726 | 6.1818 | 0.1926 | 0.2512 | 0.7863 | 0.6158 |
| **MultiBO(Ours)** | ✓ | ✓ | **0.9364** | **0.5497** | 6.6690 | 0.2266 | 0.2640 | 0.8883 | 0.6723 |

Additionally, **MultiBO** significantly outperforms **MultiBO$_{Subspace}$** on all metrics, implying that constraining the optimization to attention space instead of latent space $x_t$ is prudent to optimize quickly under $B = 50$ queries for human-in-the-loop setting. (Please refer to the Appendix B for qualitative results). Thus, the versatility of human-in-the-loop multi-choice preference coupled with constrained Bayesian Optimization is well suited to address our target alignment task.

❷ **Alignment with Preference Reward Metrics**: *Training-based methods*: Table 1 demonstrates **MultiBO**'s comparable performance across most alignment reward metrics while substantially outperforming fine-tuned models like DiffusionDPO (Wallace et al., 2024) and IterComp (Zhang et al., 2024a) on ImageReward and Aesthetic metrics, respectively. This is particularly significant considering that DiffusionDPO and IterComp (together with a reward model) are trained on massive datasets of $58,000+$ and $55,000+$ image pairs–whereas **MultiBO** reaches these results in only $B = 50$ iterations with a single user and no model training. These results confirm that reward models remain mere proxies for human judgment; by involving humans directly, we not only improve derived metrics but also more effectively bridge the gap to true target alignment.

*Inference-time methods*: Inference-time methods DNO (Tang et al., 2024), DEMON (Yeh et al., 2024), and DAS (Kim et al., 2025) maximize popular reward metrics rather than pursuing target alignment. While DNO and DEMON optimize diffusion noise and DAS employs Sequential Monte Carlo (SMC) to sample aligned distributions, **MultiBO** uses probabilistic optimization. Unlike SMC's

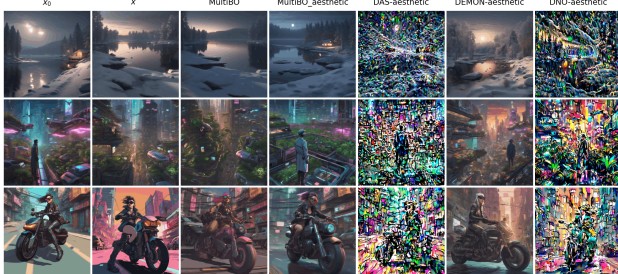

*Figure 4.* Qualitative comparison of **MultiBO** ($B = 50$), **MultiBO$_{Aesthetic}$**, DNO$_{Aesthetic}$, DEMON$_{Aesthetic}$, and DNO$_{Aesthetic}$.

focus on distribution estimation, our BO framework directly identifies the optimum $\hat{x}^*$, making it uniquely suited for the target alignment task.

For a fair comparison, alongside **MultiBO**, we evaluate $\{method\}_{<reward>}$ pairs against **MultiBO$_{<reward>}$** (where the human is replaced by the corresponding $<reward>$ model in the BO loop). The results in Table 1 and Figure 4 reveal three critical insights:

**a) Broad Applicability across Rewards:** Unsurprisingly, for any given reward, **MultiBO$_{<reward>}$**—along with DNO, DEMON, and DAS operating on that same reward—achieves peak performance for that specific metric (Table 1). **MultiBO$_{<reward>}$** shows consistent cross-reward performance, demonstrating that our optimization framework is broadly applicable and comparable to existing alignment methods in maximizing diverse objectives.

**b) Robustness to Reward Hacking:** Existing alignment methods are often prone to reward-hacking. As shown in Figure 4, DNO$_{Aesthetic}$ and DAS$_{Aesthetic}$ generate very poor images despite achieving high Aesthetic scores ($\sim 7.7$) in Table 1. In contrast, **MultiBO** avoids this pitfall, maintain-

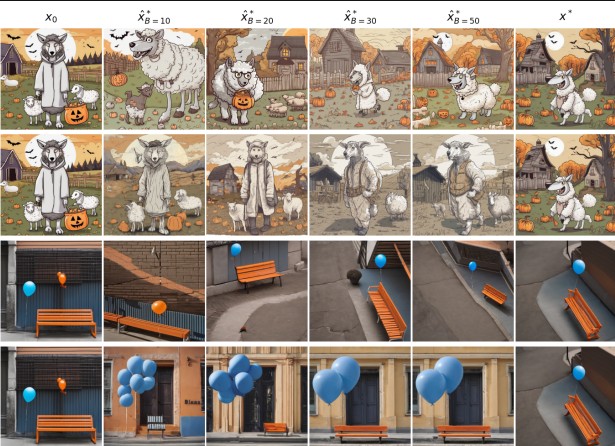

*Figure 5.* Qualitative results comparing **MultiBO** (*1st & 3rd row*) and DEMON *choose generate*(*2nd & 4th row*).

ing visual integrity where others fail.

**c) Efficiency of Human Feedback:** The most critical result is that **MultiBO** (with a human scorer), despite not explicitly optimizing for any specific proxy reward, performs comparably to DNO, DAS, and DEMON on their own target metrics. This emphasizes that high-quality user-in-the-loop preference input is superior to proxy metrics. By accounting for user burden as a key design constraint, **MultiBO** successfully extracts this high-value information to achieve robust alignment while remaining practically viable.

*Table 2.* Human evaluation using Win Rate (WR) (↑), Mean Rank (MR)(↓), MOS(↑). Kendall's W=0.65 (high inter-rater agreement).

| Method | DAS (PicScore) | DNO (PicScore) | Iter Comp | DEMON (Aesthetic) | MultiBO (Aesthetic) | DEMON (choose) | MultiBO (LPIPS) | MultiBO (Ours) |
|---|---|---|---|---|---|---|---|---|
| **WR(%)** | 0.53 | 5.36 | 6.49 | 9.55 | 11.26 | 15.60 | 31.81 | **70.82** |
| **MR** | 2.98 | 2.87 | 2.81 | 2.79 | 2.73 | 2.67 | 2.36 | **1.31** |
| **MOS** | 1.03 | 1.25 | 1.38 | 1.42 | 1.53 | 1.66 | 2.27 | **3.58** |

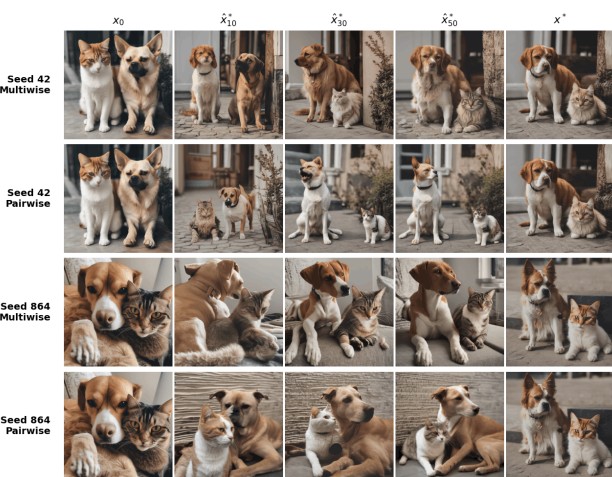

*Figure 6.* Ablation - Pairwise vs Multiwise Optimization Progress for two different seeds.

For a true analysis of **MultiBO**'s target and reward alignment, we compare against DEMON's *choose generate* mode, where user selection acts as a non-differentiable re-

ward ($+1/-1$), mirroring our PairwiseGPR formulation (Eqn. 5). Figure 5 shows that **MultiBO** converges closer to $x^*$ within 50 iterations, whereas DEMON stalls around iteration 30. This confirms that naive human-in-the-loop integration is insufficient; effective optimization must balance editing freedom with user burden, which **MultiBO** achieves via MultiwiseGPR and DBS ACF. Furthermore, unlike DEMON's high-dimensional latent noise ($\epsilon_t$) optimization, **MultiBO** operates in a lower-dimensional attention space, enabling significantly faster alignment.

■ **Human Evaluation**: We report human evaluation results from 30 volunteers. Participants were asked to pick the Top-2 methods closest to the target $x^*$. A method is ranked 1, 2 or 3 (not in top 2). Table 2 demonstrates that **MultiBO** has a high win rate compared to baselines with high user agreement. Please refer to the Appendix B for additional results including qualitative results on all other reward metrics.

■ **Ablations**: (1) *PairwiseGPR likelihood ($K = 2$) vs MultiwiseGPR ($K > 2$)*: In $B = 50$ steps, Figure 6 and Table 3 illustrate that PairwiseGPR requires far more preference feedback, $B$ to sufficiently span the optimization input space for accurately modeling $f$ and identifying it's optimum $x^*$.

(2) *MultiBO with Inconsistent User Preference*: To mimic unreliability, a fraction of user preferences are replaced with random inputs to **MultiBO**, as if the users are uniformly

| Method | CLIP-I | LPIPS |
|---|---|---|
| Pairwise | 0.81 | 0.68 |
| Multiwise | **0.94** | **0.55** |

*Table 3.* Target metrics

randomly selecting the images. This fraction varies from 10% to 75% of the iterations. Figure 7 shows that the performance is robust up to 30%. This robustness is attributed to **MultiwiseGPR**'s inherent modeling of noise (unreliability) when accepting a user's preference. Quantitatively (Table 8), **MultiBO** with inconsistent input is still comparable to other baselines in Table 1, indicating that **MultiBO** learns broad human preference despite unreliable feedback.

(3) *MultiBO with far-away starting point $x_0$*: A visually far-away starting point obviously makes the problem more challenging. This is particularly true when (1) the prompt misses some objects that should be present in the target image, or (2) when the prompt is good but the model makes mistakes (e.g., an attribute-object mismatch where the generated image has a red apple on a green bowl even though the prompt said "green apple on a red bowl"). **MultiBO** is unable to solve case (1) since bringing back missing objects is almost impossible (the space of plausible objects is extremely high and there is no guidance towards correct objects). However, we find that case (2) can often be corrected, at the expense of higher number of queries to users (i.e., higher B), as observed in Table 9 and Figure 8. The corresponding prompts are in Table 10 and more ablations are available in Appendix A.4.3

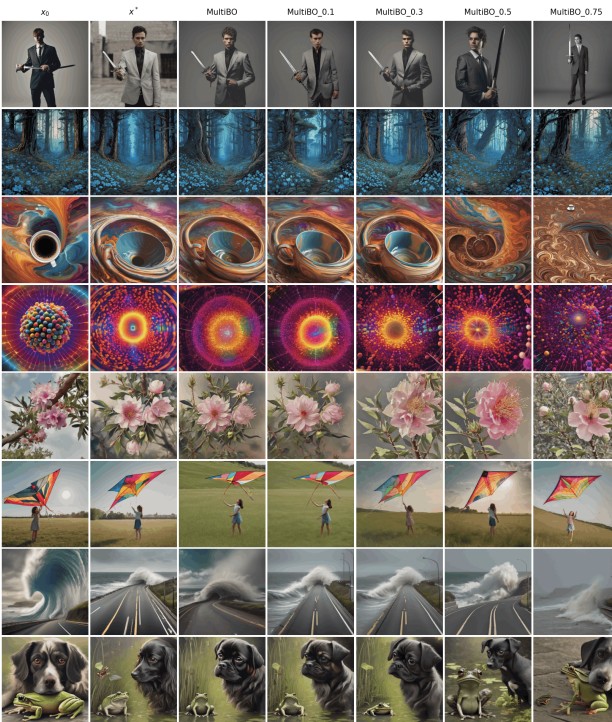

*Figure 7.* Qualitative examples with different amounts $(10, 30, 50, 75\%)$ of unreliable user preference ($B = 50$).

## 5. Related Works

■ **Preference Alignment in Diffusion Models**: Aligning diffusion with human preferences is critical for enhancing generation quality and user satisfaction. Direct Preference Optimization (DPO) methods like DiffusionDPO (Wallace et al., 2024) bypass reward model training, directly finetuning on preference data. IterComp (Zhang et al., 2024a) and CaPO (Lee et al., 2025) extend this by iteratively training rewards or calibrating preferences without annotations. To mitigate reward over-optimization, recent works use sample inversion techniques: DDIM-InPO (Lu et al., 2025a), SmPO-Diffusion (Lu et al., 2025b), InversionDPO (Li et al., 2025) reformulate DPO loss using inversion or smoothed distributions. Training-free approaches target inference time. DAS (Kim et al., 2025) uses Sequential Monte Carlo, while DNO (Tang et al., 2024) optimizes noise via reward guidance. DEMON (Yeh et al., 2024) proposes stochastic optimization to guide denoising without backpropagation. However, challenges regarding reward hacking, computational overhead, and personalization persist.

■ **Training-free Attention based Methods**: Attention-based spatial editing employs strategies like feature injection (Zhou et al., 2025; Chen & Huang, 2023; Cao et al., 2023; Huang et al., 2023a; Khandelwal, 2023; Wu et al., 2024; Long et al., 2024; Tumanyan et al., 2023), attention map operations (Wu et al., 2024; Mou et al., 2024), feature concatenation (Balaji et al., 2022; Mou et al., 2023; Deng et al., 2023), cross-modal optimization (Hertz et al., 2022;

Esser et al., 2024), and adapter-based re-learning (Ye et al., 2024). Works such as DreamMatcher (Nam et al., 2024) achieves semantic alignment via source $Q, K$ injection and $V$ warping. While effective at reducing inconsistencies, such methods lack free-form editing capabilities, typically relying on source images or custom instructions.

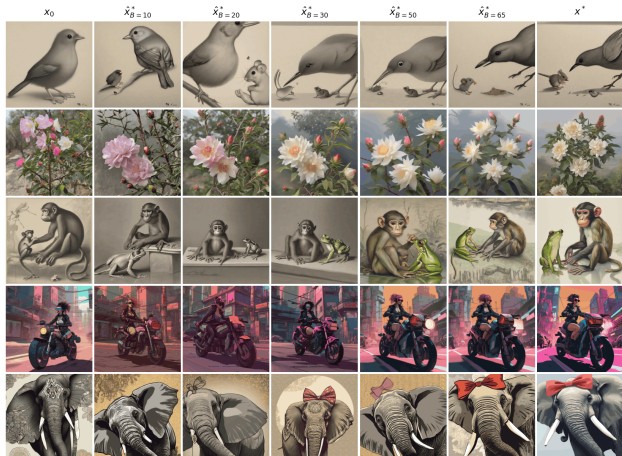

*Figure 8.* Qualitative examples of **MultiBO** for far away starting images $x_0$ (missing objects, wrong colors, etc.)

## 6. Conclusion, Limitations & Future Work

We propose **MultiBO**, a training-free, human-in-the-loop image personalization framework built upon Preferential Bayesian Optimization. **MultiBO** addresses the "last-mile" gap between a user's latent visual intent and the sub-optimal images produced by T2I models. With only 50 queries, the method converges to images that closely align with the user's ideal target, without requiring any task-specific training data. Query efficiency is further improved by incorporating multi-choice preference feedback and constraining the attention optimization to valid warping transformations, substantially reducing the search complexity.

While **MultiBO** achieves good target alignment within a budget $B$, Figure 24 highlights room for improvement. We diagnose the problem as twofold. First, there are harder scenarios where the generation is reasonably close to the target and the user is forced to choose between equally bad or good choices, i.e., the preference signal weakens, stalling the optimization. Secondly, more iterations $B > 50$ may be needed. This is influenced by the decision to optimize only in the attention domain and restricting to a transformation space, limiting the arsenal of possible changes that can be affected. For future work, we plan to address these issues by incorporating informed priors into the Gaussian likelihood of BO by leveraging pre-trained reward models. In addition, we aim to explore customized high-dimensional Bayesian optimization strategies that are better suited to the structure and geometry of latent diffusion representations.

## Impact Statement

**MultiBO** achieves hyper-personalized text-to-image generation by incorporating a human in the loop to optimize their preference. This introduces certain risks. It could be misused to create deceptive or misleading visuals, contributing to the spread of misinformation, and may compromise personal privacy. Moreover, the generated content can raise concerns around copyright and intellectual property.

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

# A. Appendix

## A.1. Gaussian Process Likelihoods & Posterior

We derive the GPR likelihood expressions for Multi-wiseGPR modeling 1-out-of-$K$ preference choices in A.1.2, and SubsetMultiwiseGPR modeling $N$-out-of-$K$ preference choices in A.1.3 respectively.

### A.1.1. PAIRWISE BINOMIAL LOGIT LIKELIHOOD GPR

The GPR posterior 7 constructed from the binomial likelihood in Eqn 5 has no closed-form expression and although there exist sophisticated variational and Monte Carlo methods, it is typically estimated by Laplace approximation (Brochu et al., 2010). The Laplace approximation follows from Taylor-expansion of the log-posterior of GPR about a set point $\hat{\mathbf{f}} = \mathbf{f}_{\mathrm{MAP}}$, the MAP estimation and is given as,

$$\log P(\mathbf{f}|\mathcal{X}) = \log P(\hat{\mathbf{f}}|\mathcal{X}) + \mathbf{g}^T(\mathbf{f} - \hat{\mathbf{f}}) \\ - \frac{1}{2}(\mathbf{f} - \hat{\mathbf{f}})^T \mathbf{H}(\mathbf{f} - \hat{\mathbf{f}}) \tag{19}$$

where, $\mathbf{g}, \mathbf{H}$ are the gradient and Hessian respectively. Their closed form expressions are found in (Chu & Ghahramani, 2005),

### A.1.2. MULTIWISE LOGIT LIKELIHOOD GPR

We derive the likelihood expression in Eqn. 10 as follows,

Consider a choice set $\mathcal{Z} = \{1, \ldots, K\}$. Each choice $j \in \mathcal{Z}$ is associated with a latent utility

$$v_j = f_j + \delta_j, \tag{20}$$

where $f_j \in \mathbb{R}$ is a true function value and $\delta_j$ is random noise. The observed choice is

$$a = \arg\max_{j \in \mathcal{Z}} v_j. \tag{21}$$

The probability that choice $i$ is chosen is

$$\begin{aligned} P(a = i \mid \mathbf{f}) &= P(v_i \geq v_j \;\; \forall j \neq i) \\ &= P(f_i + \delta_i \geq f_j + \delta_j \;\; \forall j \neq i) \\ &= P(\delta_j \leq \delta_i + f_i - f_j \;\; \forall j \neq i). \end{aligned} \tag{22}$$

$$\begin{aligned} P(a = i \mid \mathbf{f}) &= \int P(\delta_j \leq \delta_i + f_i - f_j \;\; \forall j \neq i \mid \delta_i)\, p(\delta_i)\, d\delta_i \\ &= \int \prod_{j \in \mathcal{Z},\, j \neq i} F(\delta_i + f_i - f_j)\, p(\delta_i)\, d\delta_i, \end{aligned} \tag{23}$$

where $F(\cdot)$ is the CDF of $\delta_j$.

Assume that $\delta_j \sim$ IID Gumbel$(0,1)$, with CDF and PDF,

$$F(\delta) = \exp(-e^{-\delta}), \qquad p(\delta) = \exp\left(-(\delta + e^{-\delta})\right). \tag{24}$$

Substituting them,

$$P(a = i \mid \mathbf{f}) = \int \prod_{j \in \mathcal{Z},\, j \neq i} \exp\left(-e^{-(\delta_i + f_i - f_j)}\right) \\ \exp\left(-(\delta_i + e^{-\delta_i})\right) d\delta_i. \tag{25}$$

Simplifying,

$$\prod_{j \in \mathcal{Z},\, j \neq i} \exp\left(-e^{-(\delta_i + f_i - f_j)}\right) = \exp\left(-e^{-\delta_i} e^{-f_i} \sum_{j \in \mathcal{Z},\, j \neq i} e^{f_j}\right). \tag{26}$$

Thus,

$$P(a = i \mid \mathbf{f}) = \int \exp\left(-\delta_i - e^{-\delta_i}\left(1 + e^{-f_i} \sum_{j \in \mathcal{Z},\, j \neq i} e^{f_j}\right)\right) d\delta_i. \tag{27}$$

Change of variables $t = e^{-\delta_i}$, and $d\delta_i = -dt/t$. The integral becomes,

$$P(a = i \mid \mathbf{f}) = \int_0^\infty \exp\left(-t\left(1 + e^{-f_i} \sum_{j \in \mathcal{Z},\, j \neq i} e^{f_j}\right)\right) dt. \tag{28}$$

which gives,

$$P(a = i \mid \mathbf{f}) = \frac{1}{1 + e^{-f_i} \sum_{j \in \mathcal{Z},\, j \neq i} e^{f_j}} = \frac{e^{f_i}}{\sum_{j \in \mathcal{Z}} e^{f_j}}. \tag{29}$$

Therefore, the multinomial-logit probability is the softmax function (Eqn. 10)

$$P(a = i \mid \mathbf{f}) = \frac{\exp(f_i)}{\sum_{j \in \mathcal{Z}} \exp(f_j)}. \tag{30}$$

We use Laplace Approximation (Eqn. 19) to estimate the posterior in Eqn. 12. The corresponding gradient and Hessian are,

$$\nabla \log p(\mathbf{f} \mid \mathcal{X}) = -\mathbf{K}^{-1}\mathbf{f} + \sum_{i=1}^{M} (\mathbf{e}_{a_i} - \boldsymbol{\pi}_i), \tag{31}$$

$$\nabla^2 \log p(\mathbf{f} \mid \mathcal{X}) = -\mathbf{K}^{-1} - \sum_{i=1}^{M} \left[\mathrm{diag}(\boldsymbol{\pi}_i) - \boldsymbol{\pi}_i \boldsymbol{\pi}_i^\top\right], \tag{32}$$

where, $\mathbf{K}$ is the GPR prior, $\mathbf{e}_{a_i}$ is the one-hot indicator vector for the observed choice $a_i \in \mathcal{Z}_i$, i.e., the winner

$a_i$ in $i$th choice set $\mathcal{Z}_i$ is $x_j$ from the set of $N$ samples $[x_1, \ldots, x_N]$, $(\mathbf{e}_{a_i})_j = \mathbf{1}[x_j = a_i]$. and,

$$(\pi_i)_j = P(a = x_j|\mathbf{f}) = \frac{\exp(f_j)}{\sum_{k \in \mathcal{Z}_i} \exp(f_k)} \qquad (33)$$

is the softmax function.

### A.1.3. SUBSET-CHOICE MULTIWISE LOGIT LIKELIHOOD GPR

We consider a more general form of multiwise preference scenario, where instead of selecting one choice, the user selects a *non-empty subset* of the $K$ choices.

Let $\mathcal{Y} = \{x_1, \ldots, x_N\}$ be a set of $N$ samples. A single choice set of size $K$ is,

$$\mathcal{Z} = \{a^{(1)}, \ldots, a^{(K)}\} \subseteq \mathcal{Y}.$$

The user preference is a non-empty subset

$$A \subseteq \mathcal{Z}, \qquad A \neq \emptyset.$$

The observation to the likelihood model is,

$$\mathcal{X} = \{A \succ \emptyset \mid A \subseteq \mathcal{Z}\}.$$

We assume a Gaussian prior over the latent utility function $f$,

$$f \sim \mathcal{N}(0, \mathbf{K}),$$

The latent noisy function values are,

$$\mathbf{f} = \{f(x) : x \in \mathcal{Z}\}.$$
$$v(x) = f(x) + \delta,$$

where the noise is IID $\delta \sim \text{Gumbel}(0, 1)$,

Unlike the multinomial-logit model, which assumes a single winning choice, we assume that each element in $\mathcal{Z}$ is independently selected or rejected, and the chosen subset $A$ corresponds to the selected elements.

For a *subset* $A \subseteq \mathcal{Z}$, we define the observed choice as:

$$x_j \in A, \quad v_j \text{ is "chosen".}$$

Assuming IID $\text{Gumbel}(0, 1)$ noise, the probability of observing subset $A$ is same as the multinomial-logit derivation in Eqn. 29. The likelihood of a sample $x_j$ to be in the chosen subset $x_j \in A$ is,

$$P(x_j \in A \mid \mathbf{f}) \propto \exp(f_j),$$

Assuming each sample $x_j$ in choice set $\mathcal{Z}$ is independently selected or not, to be part of subset $A$, the joint probability of all samples in $A$ is then,

$$\prod_{x_j \in A} \exp(f_j) = \exp\Big(\sum_{x_j \in A} f_j\Big).$$

The likelihood probability normalization factor (similar to denominator of Eqn. 29) is obtained by summing over all non-empty subsets of $\mathcal{Z}$ as $A$ is one such subset,

$$\sum_{\emptyset \neq C \subseteq \mathcal{Z}} \exp\Big(\sum_{x_j \in C} f_j\Big).$$

Simplifying,

$$\sum_{C \subseteq \mathcal{Z}} \prod_{x_j \in C} e^{f_j} = \prod_{x_j \in \mathcal{Z}} (1 + e^{f_j}),$$

and removing the empty set case, it becomes,

$$\prod_{x_j \in \mathcal{Z}} (1 + e^{f_j}) - 1.$$

Thus, the likelihood of observing a subset $A \subseteq \mathcal{Z}$ is,

$$P(A \mid \mathbf{f}) = \frac{\exp\Big(\sum_{x_j \in A} f_j\Big)}{\prod_{x_j \in \mathcal{Z}} (1 + e^{f_j}) - 1}. \qquad (34)$$

This likelihood reduces to the multinomial-logit model (Eqn. 10) when $|A| = 1$.

Given a collection of $M$ observations $\mathcal{X} = \{A_i, \mathcal{Z}_i\}_{i=1}^M$, the posterior distribution is

$$P(\mathbf{f} \mid \mathcal{X}) = \frac{P(\mathbf{f})}{P(\mathcal{X})} \prod_{i=1}^M P(A_i \mid \mathbf{f}_{\mathcal{Z}_i}), \qquad (35)$$

where $P(\mathbf{f})$ is the GP prior and $P(\mathcal{X}) = \int P(\mathcal{X}|\mathbf{f})P(\mathbf{f})d\mathbf{f}$.

Same as MultiwiseGPR with one choice, we use Laplace Approximation (Eqn. 19) to estimate the posterior in Eqn. 35.

We now compute the Gradient and Hessian of the posterior. For a single observation $(A, \mathcal{Z})$, we define the indicator vector $\mathbf{Y} \in \{0, 1\}^{|\mathcal{Z}|}$ as,

$$Y_j = \begin{cases} 1, & x_j \in A, \\ 0, & \text{otherwise.} \end{cases}$$

The gradient and Hessian are,

$$\nabla \log p(\mathbf{f} \mid \mathcal{X}) = -\mathbf{K}^{-1}\mathbf{f} + \sum_{i=1}^M (\mathbf{Y}_i - \boldsymbol{\pi}_i). \qquad (36)$$

$$\nabla^2 \log p(\mathbf{f} \mid \mathcal{X}) = -\mathbf{K}^{-1} - \sum_{i=1}^M \Gamma_i. \qquad (37)$$

where, $\pi_j$ are defined as,

$$
\begin{aligned}
\pi_j &= P(x_j \in A \mid \mathbf{f}) \\
&= \frac{\sum_{\emptyset \neq C \subseteq \mathcal{Z}, \, C \ni x_j} \exp\left(\sum_{x \in C} f(x)\right)}{\sum_{\emptyset \neq C \subseteq \mathcal{Z}} \exp\left(\sum_{x \in C} f(x)\right)}.
\end{aligned}
\tag{38}
$$

where, $\emptyset \neq C_i \subseteq \mathcal{Z}_i, C_i \ni x_j$ means valid non-empty subsets of $\mathcal{Z}_i$ that contain $x_j$ in them (For $M$ observations, we have $(\pi_i)_j$).

The Hessian diagonal and off-diagonal entries are,

$$
\begin{aligned}
\Gamma_{jj} &= \pi_j(1 - \pi_j), \\
\Gamma_{jk} &= \pi_{jk} - \pi_j \pi_k, \quad j \neq k,
\end{aligned}
\tag{39}
$$

where,

$$
\pi_{jk} = P(x_j, x_k \in A \mid \mathbf{f})
$$

### A.2. Additional Background

We extend the Background provided in Sec. 2 of the main paper to include preliminaries on Latent Diffusion Models (LDMs) and Attention mechanism in diffusion models that are relevant to our problem of human-in-the-loop personalization of images.

### ■ Latent Diffusion Models (LDMs)

Diffusion models generate images via a reverse denoising process. The forward process adds noise to a clean data point $x_0$ using scales $\bar{\alpha}_t = \prod_i^t \alpha_i$, yielding $x_t = \sqrt{\bar{\alpha}_t} x_0 + (\sqrt{1 - \bar{\alpha}_t})\epsilon$, where $\epsilon \sim \mathcal{N}(0, I)$ and $x_T \sim \mathcal{N}(0, I)$.

The reverse process recovers the image by modeling $p_\theta(x_{t-1}|x_t)$. This relies on estimating the clean sample $\hat{x}_0$ using Tweedie's formula and a trained denoiser $\epsilon_\theta$:

$$
\hat{x}_0 = \frac{x_t - \sqrt{1 - \bar{\alpha}_t}\,\epsilon_\theta(x_t, t)}{\sqrt{\bar{\alpha}_t}}
\tag{40}
$$

LDMs improve efficiency by running this process in a VAE-compressed latent space. They enable control (e.g., text-to-image) via conditional denoisers $\epsilon_\theta(x_t, c, t)$, where a domain encoder $\tau_\theta$ projects prompts $c$ (e.g., CLIP embeddings) for the denoiser.

### ■ Attention in Diffusion Models

The denoiser network of Diffusion models utilize attention mechanism to prioritize relevant features for consistency and control. Cross-attention aligns visual features with conditioning input $c$ (text), while self-attention models spatial dependencies and global correlations to ensure semantic integrity.

Both mechanisms project image features at time $t$ into query $(Q_t)$, key $(K_t)$, and value $(V_t)$ vectors, yielding the attention map:

$$
\text{Attention}(Q_t, K_t, V_t) = \text{Softmax}\left(\frac{Q_t K_t^T}{\sqrt{d}}\right) V_t
\tag{41}
$$

where $Q_t, K_t, V_t \in \mathbb{R}^{H \times W \times d}$ denote projections across height $H$, width $W$, and channels $d$.

$Q$ and $K$ dictate spatial attributes (Nam et al., 2024). In cross-attention, $V$ connects these to conditioning tokens, whereas in self-attention, $V$ controls non-spatial features (color, texture), enabling fine-grained generation control.

### A.3. MultiBO Algorithm

Figure 2 illustrates the Personalized Image Generation pipeline of **MultiBO**. Algorithms 1 and 2 delineate the full end-to-end personalized generation process and the DBS acquisition strategy through pseudo code.

---

**Algorithm 1** Personalized Image Generation with **MultiBO**

---

**Input:** Prompt $P$, denoiser $\epsilon_\theta$, $x_T$ image sample seed, optimization budget $B$, Edit time $t_{\text{BO}}$ and interval $\Delta$, BO hyperparameters $\gamma$, $K$ image choices $\mathcal{Z}$, User preference $g : x_w > x_{\mathcal{Z}/w}, w \in \{1, \ldots, K\}$.

**Output:** Best image sample $\hat{x}^*$ after $B$ iterations

**for** $i = 0, \ldots, B$ **do**
  **if** $i = 0$ **then**
    Initialize $N_0$ sets of $K$ random warping parameters
    $\mathcal{X}_0 = (\Theta_1^{(j)}, \Theta_1^{(j)}, \ldots, \Theta_K^{(j)}), j = 0, \ldots, N_0$

  **else**
    $\Theta_{1:K}^{(i)} = \text{DBS}(g, P(\mathbf{f}|\mathcal{X}_i))$ ;   // Next set of choices
    $\mathcal{X}_i = [\mathcal{X}_{i-1}, \Theta_{1:K}^{(i)}]$ ;   // Aggregated data

  **Self Attention Editing:**
  **for** $t = t_{BO}, t_{BO} + \Delta$ **do**
    $Qs_t, Ks_t, Vs_t \leftarrow \epsilon_\theta(x_t, t, P)$
    $\{Qs_t', Ks_t', Vs_t'\}_{1:K} = \text{Warp}(Qs_t, Ks_t, Vs_t; \Theta_{1:K}^{(i)})$
    $\epsilon_{1:K} = \epsilon_\theta(x_t, t, P; \{Qs_t', Ks_t', Vs_t'\}_{1:K})$
    $\{x_0\}_{1:K} = \text{Sampler}(x_t, t, P, \epsilon_{1:K})$

  $g : x_{0,w} > x_{0,\mathcal{Z}/w}$ ;   // Obtain User preference choice
  Fit GPR Posterior $P(\mathbf{f}|\mathcal{X}_i)$ (Eqn. 12)

$Qs_t^*, Ks_t^*, Vs_t^* = \text{Warp}(Qs_t, Ks_t, Vs_t; \Theta^*)$ ;  // Best params
,
$\epsilon^* = \epsilon_\theta(x_t, t, P; Qs_t^*, Ks_t^*, Vs_t^*)$,
Best Image: $\hat{x}^* = \text{Sampler}(x_t, t, P, \epsilon^*)$

---

**Algorithm 2** Dynamic Balanced Subspace (DBS) ACF

**Input:** BO iteration $i$; observed data $\mathcal{X}_i$; Current GPR posterior $P(\mathbf{f}|\mathcal{X}_i)$; Current best sample $\hat{\Theta}^*$; Warp parameter space $\mathcal{Y} \subseteq \mathbb{R}^D$

**Output:** $\Theta^+_{0:K-1}$ - next set of $K$ warping parameter choices

Expected Improvement sample: $x^+ = \text{EI}(P(\mathbf{f}|\mathcal{X}_i|\mathcal{Y}))$

Bridge points:
$$\Theta_{bridge,i} = \hat{\Theta}^* + \gamma_i(\Theta^+ - \hat{\Theta}^*), \gamma_i \in [0,1]$$

---

**Dynamic Subspace Selection:**

- Compute gradient matrix $\mathbb{C}$ from MultiwiseGP posterior mean $\mu(\mathbf{x})$ near $\hat{\Theta}^*$: $\hat{\Theta} = \hat{\Theta}^* + \eta$

$$\mathbb{C} = \frac{1}{N}\sum_{i=1}^{N} \nabla\mu(\hat{\Theta})\nabla\mu(\hat{\Theta})^T$$

Eigen decomposition: $\lambda, \mathbf{u} = \text{EVD}[\mathbb{C}]$

Target dimensions: $d = \arg\max_j \frac{\lambda_j}{\lambda_{j+1}}$ $j \in \{1, 2, \ldots, D\}$ (Spectral Gap)

---

$K$ choices: $\delta_i = \sigma \sum_{j=1}^{d} \alpha_{i,j}\sqrt{\lambda_j}\mathbf{u}_j$ $\quad \alpha_{i,j} \in [-1,1]$

$\Theta^+_i = \text{proj}_{\mathcal{Y}}(\Theta_{bridge,i} + \delta_i), \quad i = 0, \ldots, K-1$

## A.4. Implementation & Evaluation Details

### A.4.1. IMPLEMENTATION DETAILS

***Diffusion Parameters***: We implement **MultiBO** on SDXL (Podell et al., 2023) with DDIM for 50 inference steps and guidance scale $= 5.0$. **MultiBO** performs attention editing during the first 20% of the denoising steps. We target the middle to later attention layers (the decoder layers in the case of UNets) as they have been shown empirically in (Liu et al., 2024) to produce significant spatial changes without the loss of consistency. The early attention layers do not hold enough semantic structure to cause precise change, often resulting in complete loss of similarity to the source while the very last layers only result in minute peripheral changes. Empirically proved in A.4.3.

***BO Parameters***: **MultiBO**'s optimization is performed for $B = 50$ iterations. $N_0 = 10$ initial Sobol samples start the optimization. We use Q-NoisyNegativeLogLikelihoodExpectedImprovement (QNoisyNEI), number of restarts $= 50$, and raw samples $= 4096$ for finding $x_{\text{EI}}$.

In DBS ACF, a spectral gap threshold of $\gamma = 2.0$ is chosen to identify subspace dimension $d$ (Eqn. 16). Nine grid points define the TPS transformation (Truong et al., 2021). The maximum limits for affine parameters are, translation $= 0.75$ (75% pixel shift), scaling factor $= 4$ (scaled up to 4 times or scaled down to $1/4^{\text{th}}$), and shear and rotation angles are limited to $+/-\pi/3$.

We use $K = 4$ preference choice set. As number of choices increases, the reliability of user preference drops as they

have too much choice, this is reflected in poor metrics reported in Table 5 for $K = 10$. $K = 6$ only fetches marginal gains disproportionate to user burden.

We run **MultiBO** for three different seeded trials per prompt/target pair. We ran all experiments on a single NVIDIA A6000 GPU (48GB).

***Datasets***: In addition to the datasets mentioned in the main text, our prompts and corresponding target images $x^*$ are curated from the extensive set of popular prompt datasets in diffusion image editing space like AttendExcite (Chefer et al., 2023), T2ICompBench (Huang et al., 2023b), RareBench (Park et al., 2025) and preference alignment space such as HPSv2 (Wu et al., 2023) benchmark dataset, GenEval (Ghosh et al., 2023), PartiPrompts (Yu et al., 2022), Pick-a-Pic (Kirstain et al., 2023), and Dalle (Zhang et al., 2024a).

### A.4.2. PROMPTS FOR QUALITATIVE RESULTS IN MAIN PAPER

| Row | Text Prompt |
|---|---|
| *Prompts for Figure 3 (**MultiBO** Progress)* | |
| P1 | "A person in a suit holding a sword." |
| P2 | "A forest with blue flowers illustrated in a digital matte style by Dan Mumford and M.W Kaluta." |
| P3 | "A swirling, multicolored portal emerges from the depths of an ocean of coffee, with waves of the rich liquid gently rippling outward. The portal engulfs a coffee cup, which serves as a gateway to a fantastical dimension. The surrounding digital art landscape reflects the colors of the portal, creating an alluring scene of endless possibilities." |
| P4 | "an electron cloud model is displayed in vibrant colors with a light spectrum background, showcasing the probability distribution of electrons around the nucleus. the image resembles digital art with pixelated elements, bringing a modern, educational twist to atomic structure visualization." |
| P5 | "The fragrant flowers bloomed on the sturdy stem and the thorny bush." |
| *Prompts for Figure 4 (Aesthetic Metric)* | |
| P6 | "A vividly realistic depiction of a snowy Swedish lake at night with hyper-detailed, cinematic-level artistry showcased on ArtStation." |
| P7 | "On the rooftop of a skyscraper in a bustling cyberpunk city, a figure in a trench coat and neon-lit visor stands amidst a garden of bio-luminescent plants, overlooking the maze of flying cars and towering holograms. Robotic birds flit among the foliage, digital billboards flash advertisements in the distance." |
| P8 | "A cyberpunk woman on a motorbike drives away down a street while wearing sunglasses." |
| *Prompts for Figure 5 (DEMON choose generate)* | |
| P9 | "A wolf wearing a sheep halloween costume going trick-or-treating at the farm" |
| P10 | "a blue balloon and a orange bench" |

*Table 4.* Text prompts for figures in main text.

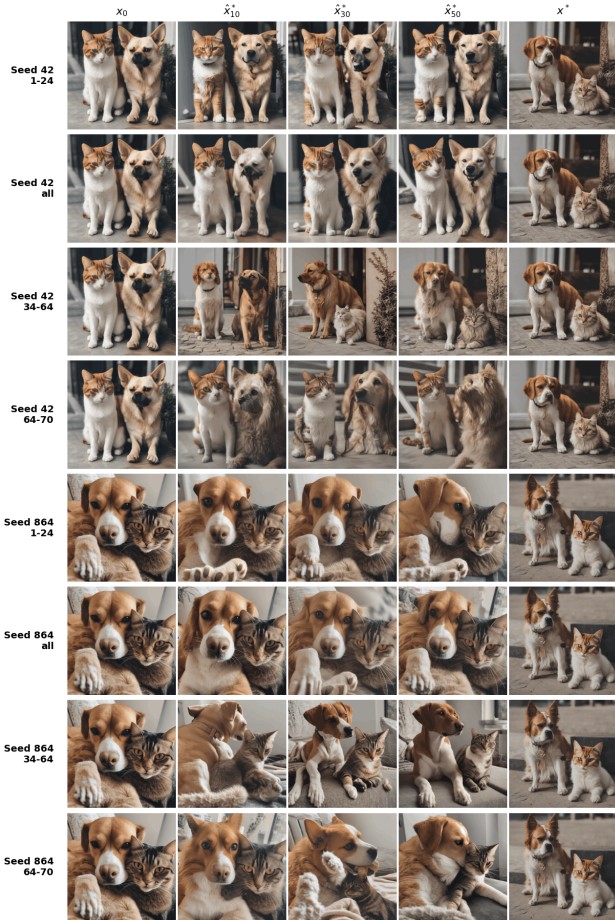

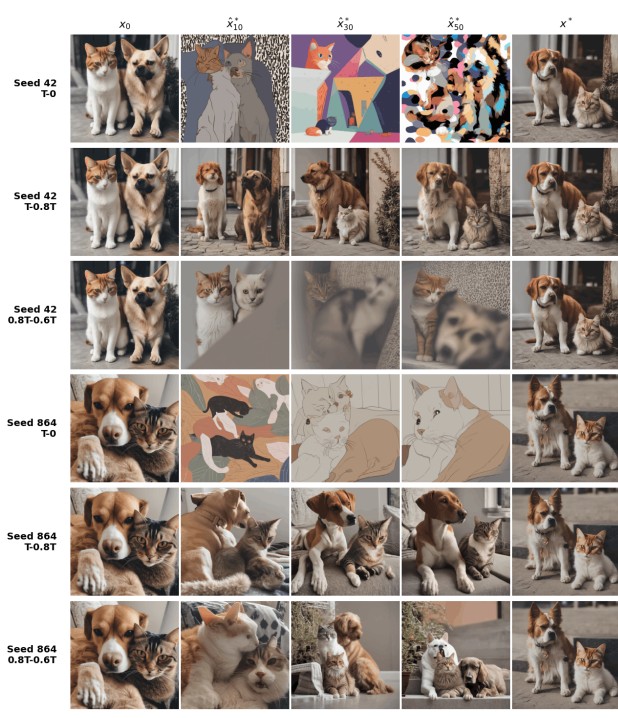

*Figure 10.* Ablation - **MultiBO** applied at different timesteps $t$ and intervals $\Delta$. (Attn layers $34 - 64$, MultiwiseGPR)

*Figure 9.* Ablation - **MultiBO** applied to different attention layers two different seeds. ($t^+ = 0.0, \Delta = 0.2$, MultiwiseGPR. For prompt: *"a cat and a dog"*)

### A.4.3. ADDITIONAL ABLATION STUDIES

Apart from the performance comparison of PairwiseGPR and MultiwiseGPR for $B = 50$ iterations, we also consider other relevant ablations,

(1) *Number of choices $K$ in choice set $\mathcal{Z}$*: As number of choices increases, the reliability of user preference drops as they have too much choice, this is reflected in poor metrics reported in Table 5 for $K = 10$. $K = 6$ only fetches marginal gains disproportionate to user burden. $K = 4$ offers good trade-off between informative preference data and minimal user burden.

(2) *Self-Attention Layers to Edit*: Middle to later self-attention layers (decoder layers of UNet) are ideal for spatial and semantic control. The early attention layers do not hold enough semantic structure to cause precise change, often resulting in complete loss of similarity to the source image while the very last layers only result in minute peripheral changes. Figure 9 shows the complete loss of semantic structure when $1 - 24$ attention layers of SDXL model are modified and the lack of significant changes when editing layers $64 - 70$. Layers $34 - 64$ offer the most editing advantage.

(3) *Timesteps to Edit*: Figure 10 identifies that the earlier ($T$ to $0.8T$) we perform edits the more freedom we have in steering the diffusion generation. Editing attention should not be done for prolonged number of timesteps ($T$ to $0$) as it steers the diffusion process to low probability points.

| K | LPIPS |
|----|-------|
| 4 | 0.55 |
| 6 | 0.54 |
| 10 | 0.64 |

*Table 5.* **MultiBO** using $K$ preference choices in MultwiseGPR

(4) *Memory and Time Analysis*:
**MultiBO** trades off latency for hyper-personalization, opening a new capability in image generation. Tables 7 and 6 report results from memory and timing analysis of the end-to-end **MultiBO** workflow. Below are salient points,

*Table 6.* Human feedback time across different iterations (optimization progress).

| Iterations | Feedback time per iter (s) |
|---|---|
| B = 5 | 40.88 |
| B = 10 | 33.99 |
| B = 20 | 28.06 |
| B = 30 | 24.62 |
| B = 40 | 20.01 |
| B = 50 | 18.89 |

- **MultiBO** is lightweight (same memory usage as base SDXL and negligible processing times). It converges in about $B = 32$ on average. DEMON reports a lower B because it converges quickly to a suboptimal image (not close to , the worse LPIPS and CLIP-I scores highlight this). Fig. 5 (main paper) also shows this.

- Moreover, we observe that the user feedback time reduces as B increases. The user becomes more confident in what to look for in choices, and as **MultiBO** converges, the outliers to reject appear prominent and easily identifiable (see Table 6).

- Around 65% of images converge in 35 iterations or less (see Figure 11(c)). The average wall clock time is about 20 mins, comparable to DEMON. Thus, by expending 15-20 mins the user can customize the generation to $x^*$. **MultiBO**$_{<\text{LPIPS}>}$ is much faster because it uses LPIPS metric as a proxy for human choice, taking $x^*$ as input. Fig. 18 shows that even with knowledge of $x^*$, its results fall short due to limitations of LPIPS metric.

*Table 8.* Impact of unreliable user preference on **MultiBO** performance ($B = 50$).

| Method | LPIPS ($\downarrow$) | CLIP-I ($\uparrow$) |
|---|---|---|
| MultiBO | 0.5497 | 0.9364 |
| MultiBO w/ rand pref = 10% of iters | 0.5467 | 0.9260 |
| MultiBO w/ rand pref = 30% of iters | 0.5684 | 0.9101 |
| MultiBO w/ rand pref = 50% of iters | 0.6183 | 0.8772 |
| MultiBO w/ rand pref = 75% of iters | 0.6491 | 0.8673 |

*Table 9.* Quantitave metrics LPIPS and CLIP-I computed between target $x^*$ and starting point $x_0$, results $\hat{x}^*$ after $B = 50$, and $B = 65$ iterations respectively of **MultiBO**, and baselines: **MultiBO**$_{<\text{LPIPS}>}$ and DEMON *choose generate*.

| | Initial ($x_0$) | | $\hat{x}^*_{B=50}$ | | $\hat{x}^*_{B=65}$ | |
|---|---|---|---|---|---|---|
| Method | LPIPS ($\downarrow$) | CLIP-I ($\uparrow$) | LPIPS ($\downarrow$) | CLIP-I ($\uparrow$) | LPIPS ($\downarrow$) | CLIP-I ($\uparrow$) |
| MultiBO | 0.7089 | 0.8538 | 0.5993 | 0.9151 | 0.5752 | 0.9222 |
| **MultiBO**$_{<\text{LPIPS}>}$ | 0.7089 | 0.8538 | 0.6714 | 0.8703 | 0.6477 | 0.8793 |
| DEMON | 0.7089 | 0.8538 | 0.6806 | 0.8667 | 0.6745 | 0.8746 |

| Row | Text Prompt |
|---|---|
| *Prompts for Figure 7 (Unreliable preference input)* | |
| P1 | "A person in a suit holding a sword." |
| P2 | "A forest with blue flowers illustrated in a digital matte style by Dan Mumford and M.W Kaluta." |
| P3 | "A swirling, multicolored portal emerges from the depths of an ocean of coffee, with waves of the rich liquid gently rippling outward. The portal engulfs a coffee cup, which serves as a gateway to a fantastical dimension. The surrounding digital art landscape reflects the colors of the portal, creating an alluring scene of endless possibilities." |
| P4 | "an electron cloud model is displayed in vibrant colors with a light spectrum background, showcasing the probability distribution of electrons around the nucleus. the image resembles digital art with pixelated elements, bringing a modern, educational twist to atomic structure visualization." |
| P5 | "The fragrant flowers bloomed on the sturdy stem and the thorny bush." |
| P6 | "A girl is holding a large kite on a grassy field." |
| P7 | "a tidal wave approaching a coastal road." |
| P8 | "a dog and a frog." |
| *Prompts for Figure 8 (Far-away)* | |
| P9 | "a bird and a mouse" |
| P10 | "The fragrant flowers bloomed on the sturdy stem and the thorny bush." |
| P11 | "a monkey and a frog" |
| P12 | "A cyberpunk woman on a motorbike drives away down a street while wearing sunglasses." |
| P13 | "a elephant with a bow". |

*Table 10.* Text prompts for ablation figures in main text.

(5) **MultiBO** *Performance under different Randomizations*: MultiBO is sensitive to how well the prompt covers all objects in the desired and how well the model is able to capture all objects and attributes in the prompt. If these conditions are well satisfied, then MultiBO is not too sensitive to seeds.

Figures 13, 14, and 15 show qualitative results on different randomization scenarios to highlight MultiBO's robustness to seeds, and the corresponding quantitative metrics are reported in Table 1 in the main paper. These together subsume all randomization scenarios for different prompts from the dataset. These figures highlight the robustness of MultiBO in personalizing to a specific target despite the starting point (within reason) and vice versa.

### A.5. Additional Related Works

We provide a more detailed overview of existing related works in both reward-based preference alignment space and the guidance-based diffusion generation space.

■ **RLHF and Reward Based Methods**: RLHF (Bai et al., 2022; Ouyang et al., 2022) has been extensively adapted for T2I diffusion (Domingo-Enrich et al., 2025; Jiang et al.,

*Table 7.* Memory and time analysis of **MultiBO**, and baselines: **MultiBO**$_{<LPIPS>}$ and DEMON *choose generate*. Base SDXL model memory = 8.5 GB. We report 5 sets of results: (1) quantitative metrics, (2) memory usage of different methods, (3) average per iteration image generation time, BO processing time, human feedback time, and the combined time, (4) total generation time, human feedback time, and combined time until convergence, and (5) average number of iterations and corresponding images to convergence. *MultiBO and the baselines are assumed to have converged if the resulting image $\hat{x}^*$ of two successive iterations have a difference in LPIPS score $< 10^{-2}$.*

| Method | Metrics | | Memory | Time per Iteration (s) | | | | Time until Convergence (mins) | | | Until Convergence | |
| | LPIPS | CLIP-I | GB | Gen. | BO | Human | Total | Gen. | Human | Total | Imgs. | B |
|---|---|---|---|---|---|---|---|---|---|---|---|---|
| MultiBO | 0.5497 | 0.9364 | 9.00 | 10.35 | 0.01 | 27.64 | 37.82 | 5.55 | 14.83 | 20.28 | 128 | 32.19 |
| **MultiBO**$_{<LPIPS>}$ | 0.5924 | 0.9114 | 9.14 | 10.35 | 0.01 | 0.01 | 10.36 | 6.01 | 0.01 | 7.90 | 183 | 45.73 |
| DEMON | 0.6495 | 0.8895 | 25.20 | 20.11 | NA | 34.08 | 54.19 | 7.14 | 11.84 | 18.83 | 84 | 20.85 |

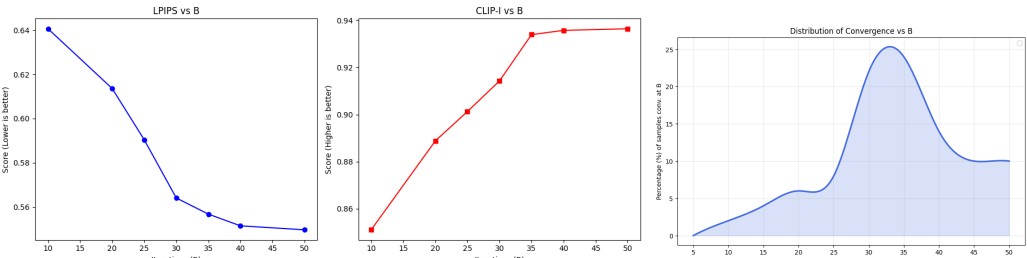

*Figure 11.* Performance of **MultiBO** on metrics LPIPS (a) and CLIP-I (b) as optimization progresses. Distribution of samples that need $B$ iterations to converge.

2024; Li et al., 2024; Uehara et al., 2024; Zhang et al., 2024b; Lee et al., 2023; Sun et al., 2023). Methods like ImageReward (Xu et al., 2023; Clark et al., 2023; Prabhudesai et al., 2023) use supervised loss with trained reward models. Others like DDPO (Black et al., 2024) and DPOK (Fan et al., 2023; 2024) formulate sampling as a Markov decision process, applying RL to maximize rewards.

■ **Guidance-based Methods**: Many guidance methods build on the score-based formulation of diffusion models (Song et al., 2021). Classifier Guidance (Dhariwal & Nichol, 2021) requires additional training, works like Universal Guided Diffusion (He et al., 2024; Yoon et al., 2023; Song et al., 2023; Yu et al., 2023; Serrà et al., 2023; Chung et al., 2023) approximate guidance to use off-the-shelf classifiers or models directly. These methods use Tweedie's formula operate on the predicted clean sample However, such methods suffer from poor guidance, and the Tweedie mean prediction quality limits effectiveness in maximizing complex rewards. *Training-free Spatial Editing methods:* InstanceDiffusion (Wang et al., 2024) uses instance-level anchors for specialized object placement and scene construction. Works like (Xie et al., 2023; Phung et al., 2024; Kim et al., 2023; Zhao et al., 2023) use specialized conditioning modalities like bounding boxes, vectors, masks, box-text pairs, custom instructions, etc. to explicitly steer the diffusion generation. These methods place coonsiderable burden on the user to provide these inputs and often perform poorly when a combination of changes are required or when the user is unable to express their requirement. Works like (Yang et al., 2024; Hu et al., 2024) (Yang et al., 2024); (Qu et al., 2023), and (Yu et al., 2025) use Mixture

of Experts (MoEs) or LLMs to parse the user requirement into several sub-editing tasks assigned to a specific editing pipeline that excels in that task. Although these methods offer an interactive editing framework they are often limited by the user's capability to express the required changes via syntax like language.

## B. More Qualitative Results

We show qualitative results for **MultiBO**, **MultiBO**$_{<reward>}$, training methods like DiffusionDPO (Wallace et al., 2024) and IterComp (Zhang et al., 2024a) and inference time methods, DNO (Tang et al., 2024), DAS (Kim et al., 2025), DEMON (Yeh et al., 2024) operating on different $<$ reward $>$ metrics. These results complement the quantitative counterparts reported in Table 1 in the main paper.

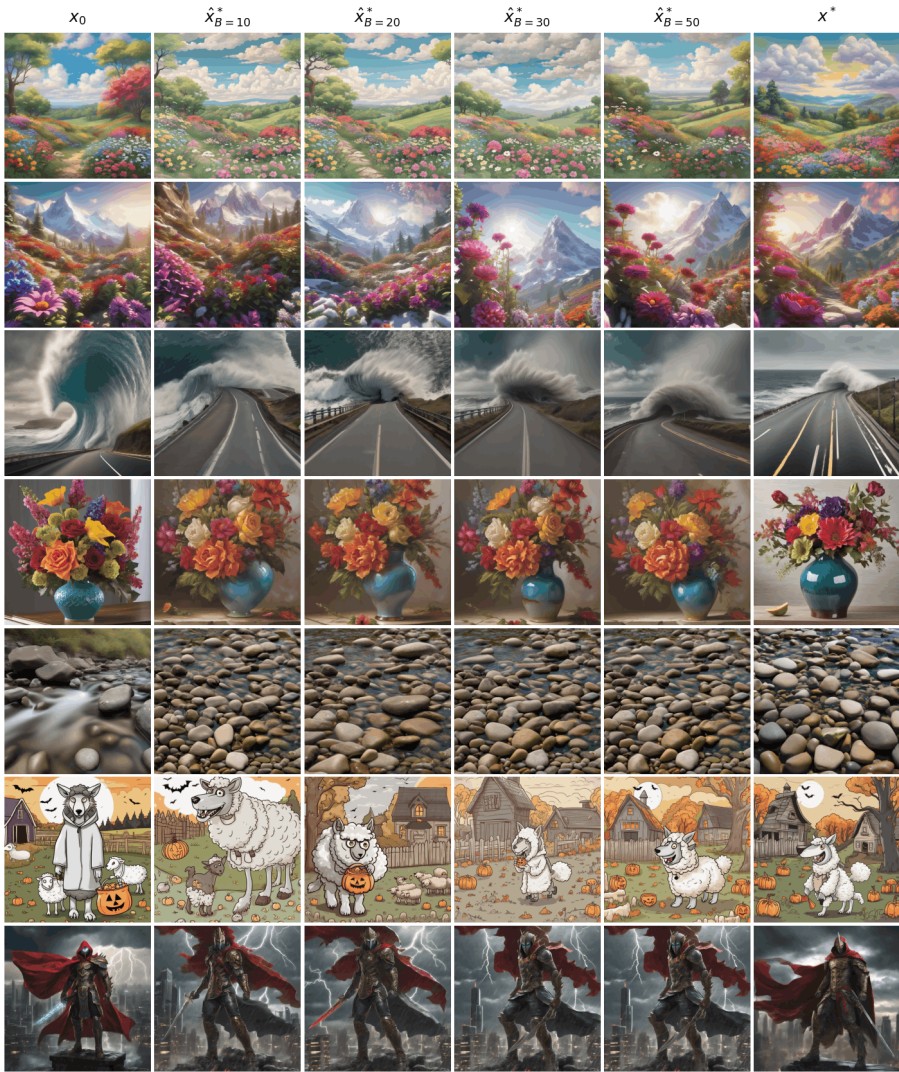

*Figure 12.* Qualitative Examples when **MultiBO** converges in $B < 50$, mostly converging around $B = 20 - 30$. For prompts: *"A peaceful, nature-filled landscape with vibrant flowers and trees and a serene cloud-filled sky.", "a stunning 3d render of towering, giant blooming plants with vibrant, colorful flowers on a picturesque mountain landscape. sunlight dances on the petals, creating an enchanting scene as the wind gently sways the plants, with snow-capped peaks in the distance.", "a tidal wave approaching a coastal road", "The smooth, glossy finish of the ceramic vase accentuated the vibrant colors of the flowers, a stunning centerpiece of beauty.", "The smooth, cool surface of the river rocks were perfect for skipping across the water's surface.", "A wolf wearing a sheep halloween costume going trick-or-treating at the farm", "Amidst a stormy, apocalyptic skyline, a masked warrior stands resolute, adorned in intricate armor and a flowing cape. Lightning illuminates the dark clouds behind him, highlighting his steely determination. With a futuristic city in ruins at his back and a red sword in hand, he embodies the fusion of ancient valor and advanced technology, ready to face the chaos ahead."*

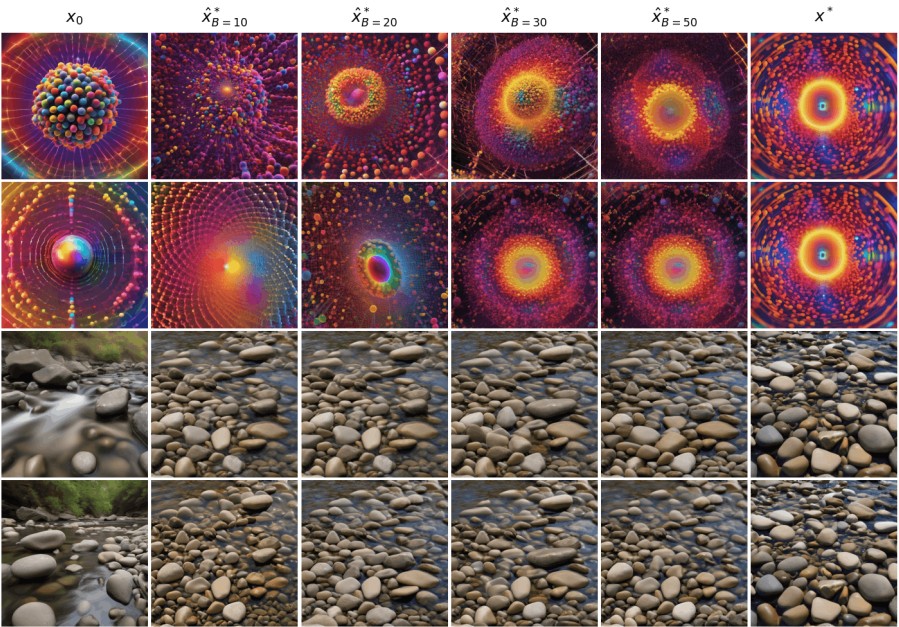

*Figure 13.* Qualitative Examples for different starts $x_0$ and same target $x^*$. For prompts: *""an electron cloud model is displayed in vibrant colors with a light spectrum background, showcasing the probability distribution of electrons around the nucleus. the image resembles digital art with pixelated elements, bringing a modern, educational twist to atomic structure visualization.","The smooth, cool surface of the river rocks were perfect for skipping across the water's surface.".*

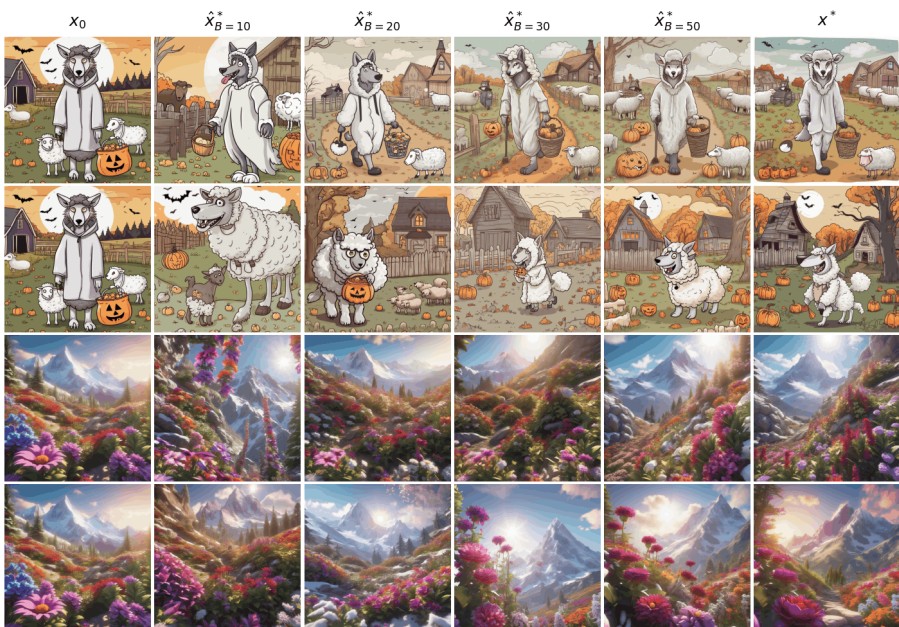

*Figure 14.* Qualitative Examples for same start $x_0$ and different targets $x^*$:. For prompts: *"A wolf wearing a sheep halloween costume going trick-or-treating at the farm", "a stunning 3d render of towering, giant blooming plants with vibrant, colorful flowers on a picturesque mountain landscape. sunlight dances on the petals, creating an enchanting scene as the wind gently sways the plants, with snow-capped peaks in the distance.".*

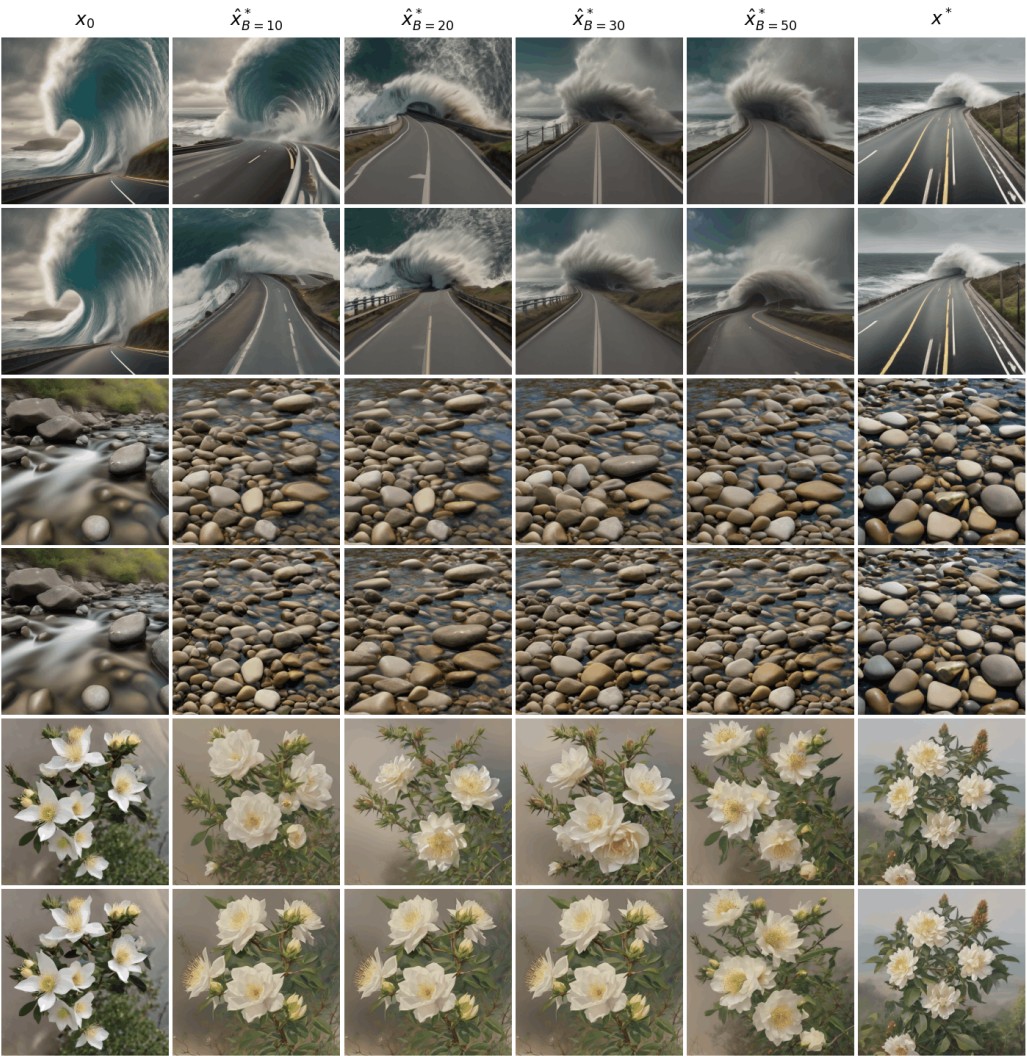

*Figure 15.* Qualitative examples for same start $x_0$ and same target $x^*$ for two **MultiBO** runs with different optimization seeds. For prompts: *"a tidal wave approaching a coastal road"*, *"The smooth, cool surface of the river rocks were perfect for skipping across the water's surface."*, *"The fragrant flowers bloomed on the sturdy stem and the thorny bush."*.

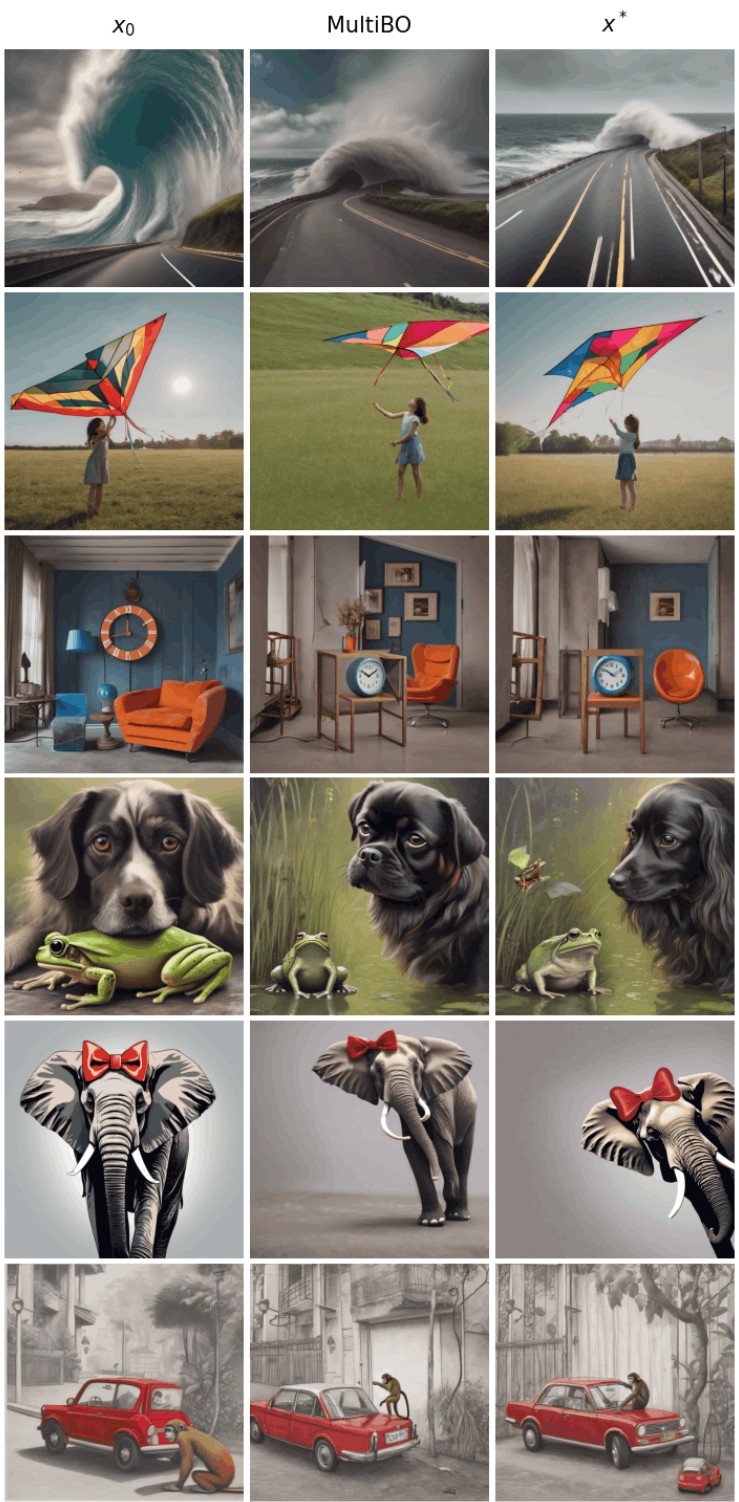

*Figure 16.* Qualitative results of **MultiBO** ($B = 50$). For prompts: *"a tidal wave approaching a coastal road","A girl is holding a large kite on a grassy field.","a orange chair and a blue clock","a dog and a frog","a elephant and a bow","a monkey and a red car"*.

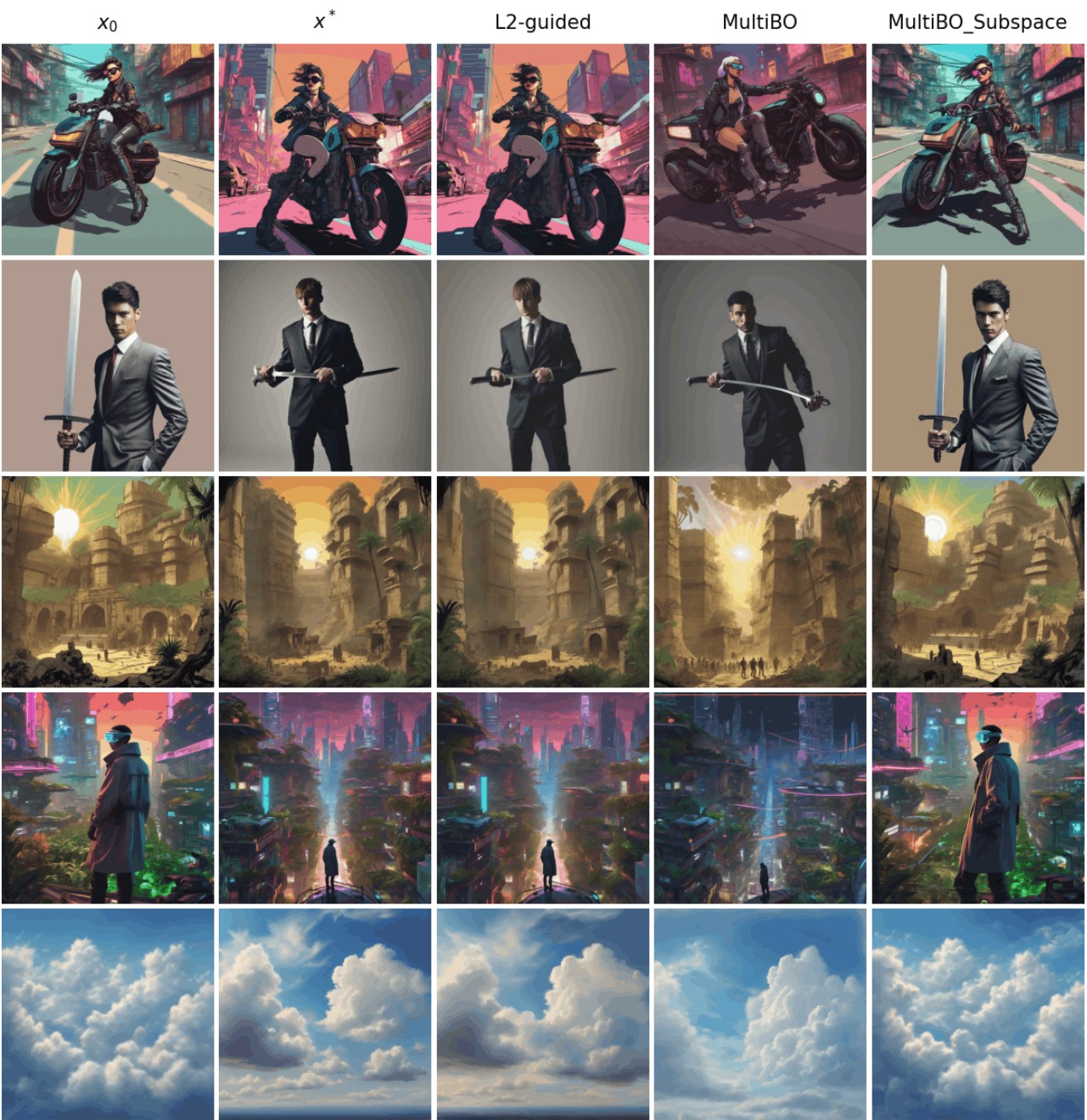

*Figure 17.* Qualitative results comparing **MultiBO** ($B = 50$), L2-guided, and **MultiBO**Subspace. For prompts: *"A cyberpunk woman on a motorbike drives away down a street while wearing sunglasses.", "A person in a suit holding a sword.", "A comic book illustration by John Kirby depicting a jungle fortress surrounded by dirt walls in a marketplace setting with cinematic rays of sunlight.", "On the rooftop of a skyscraper in a bustling cyberpunk city, a figure in a trench coat and neon-lit visor stands amidst a garden of bio-luminescent plants, overlooking the maze of flying cars and towering holograms. Robotic birds flit among the foliage, digital billboards flash advertisements in the distance.", "The soft, fluffy clouds drifted lazily across the blue sky, a canvas of endless possibilities and imagination."*

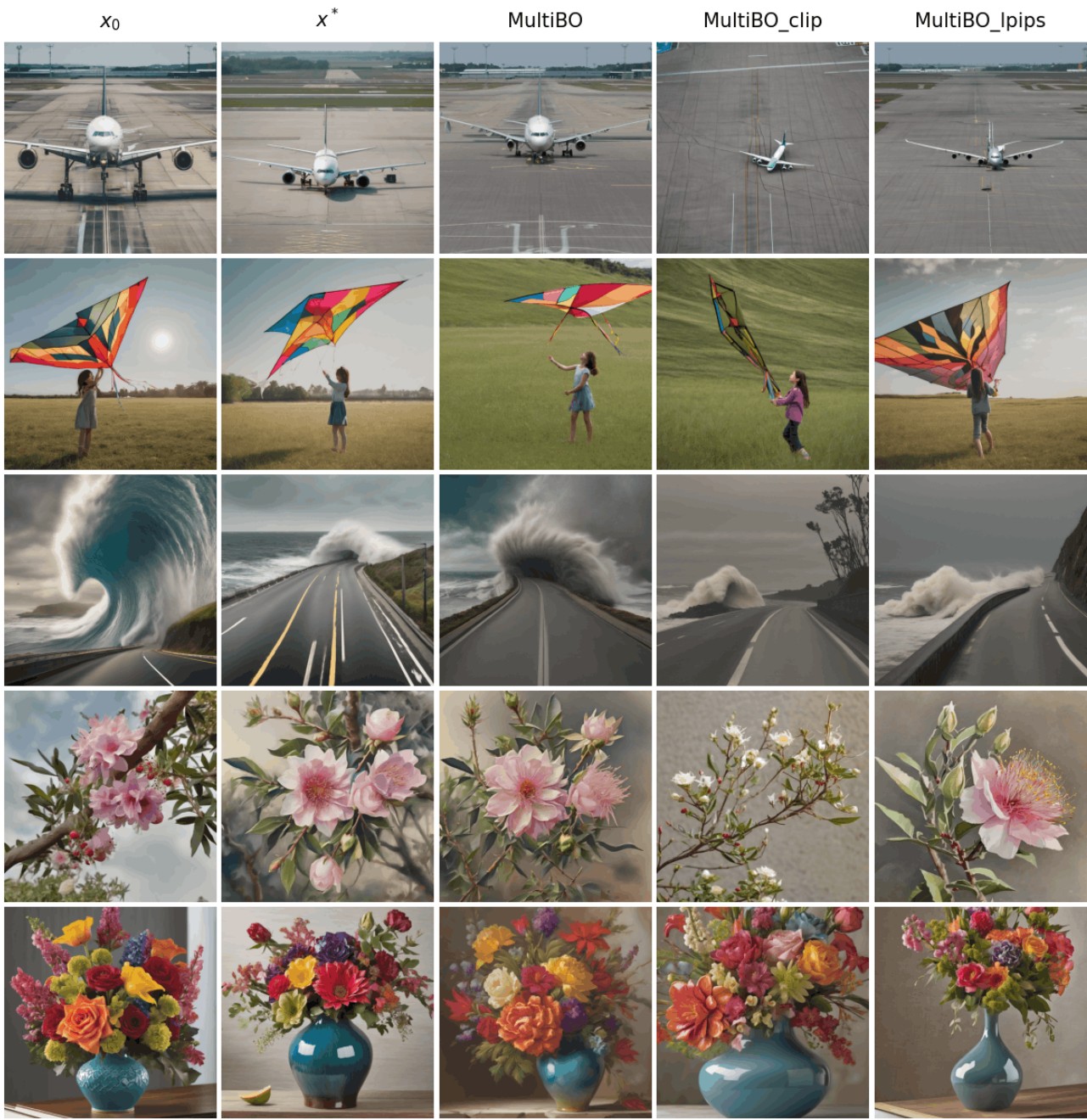

*Figure 18.* Qualitative results comparing **MultiBO** ($B = 50$), **MultiBO**$_{\text{CLIP-I}}$, and **MultiBO**$_{\text{LPIPS}}$. For prompts: *"An airplane on the runway of an airport.", "A girl is holding a large kite on a grassy field.", "a tidal wave approaching a coastal road", "The fragrant flowers bloomed on the sturdy stem and the thorny bush.", "The smooth, glossy finish of the ceramic vase accentuated the vibrant colors of the flowers, a stunning centerpiece of beauty."*

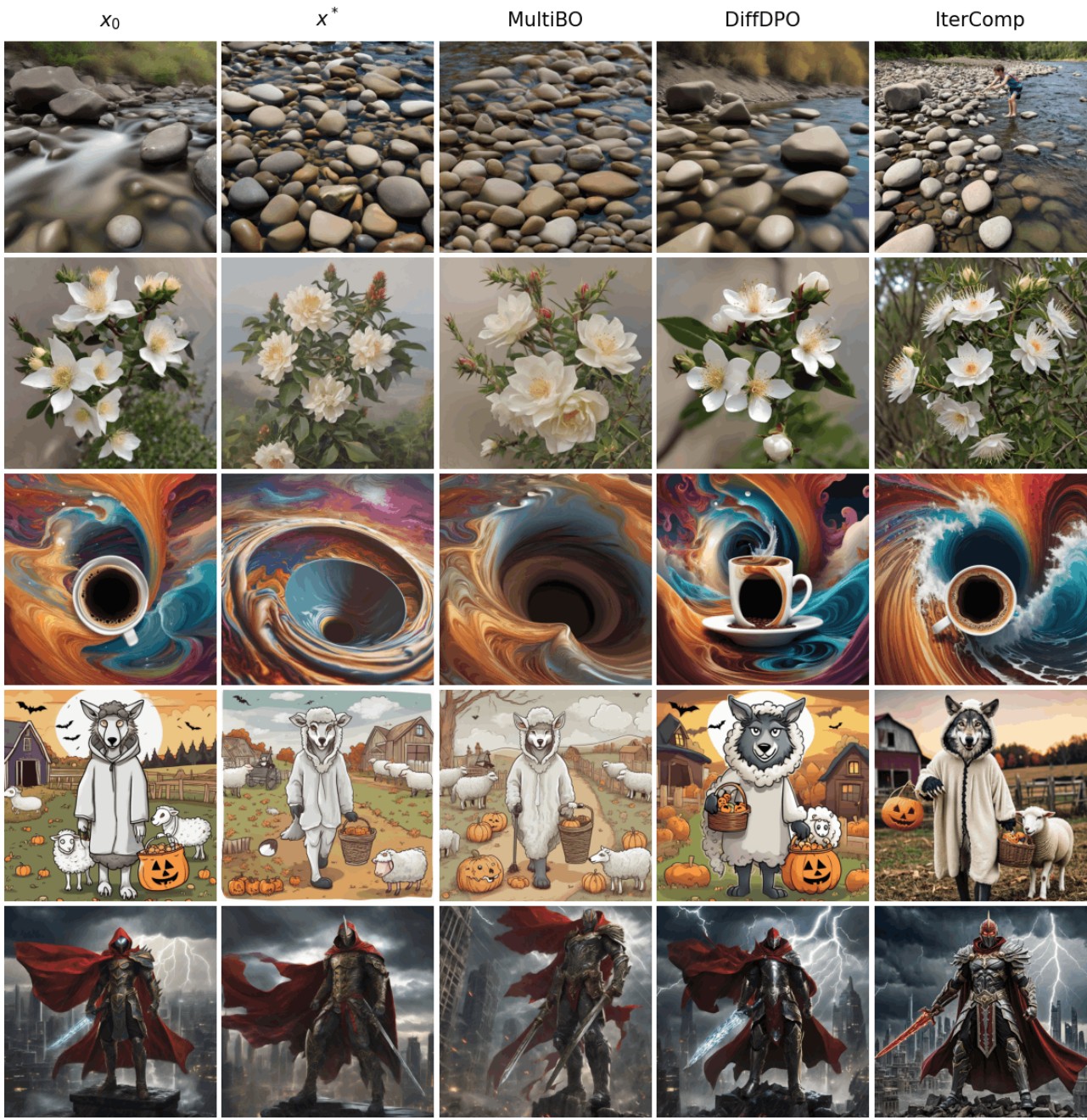

*Figure 19.* Qualitative results comparing **MultiBO** ($B = 50$), DiffusionDPO, and IterComp. For prompts: *"The smooth, cool surface of the river rocks were perfect for skipping across the water's surface.", "The fragrant flowers bloomed on the sturdy stem and the thorny bush.", "A swirling, multicolored portal emerges from the depths of an ocean of coffee, with waves of the rich liquid gently rippling outward. The portal engulfs a coffee cup, which serves as a gateway to a fantastical dimension. The surrounding digital art landscape reflects the colors of the portal, creating an alluring scene of endless possibilities.", "A wolf wearing a sheep halloween costume going trick-or-treating at the farm", "Amidst a stormy, apocalyptic skyline, a masked warrior stands resolute, adorned in intricate armor and a flowing cape. Lightning illuminates the dark clouds behind him, highlighting his steely determination. With a futuristic city in ruins at his back and a red sword in hand, he embodies the fusion of ancient valor and advanced technology, ready to face the chaos ahead."*

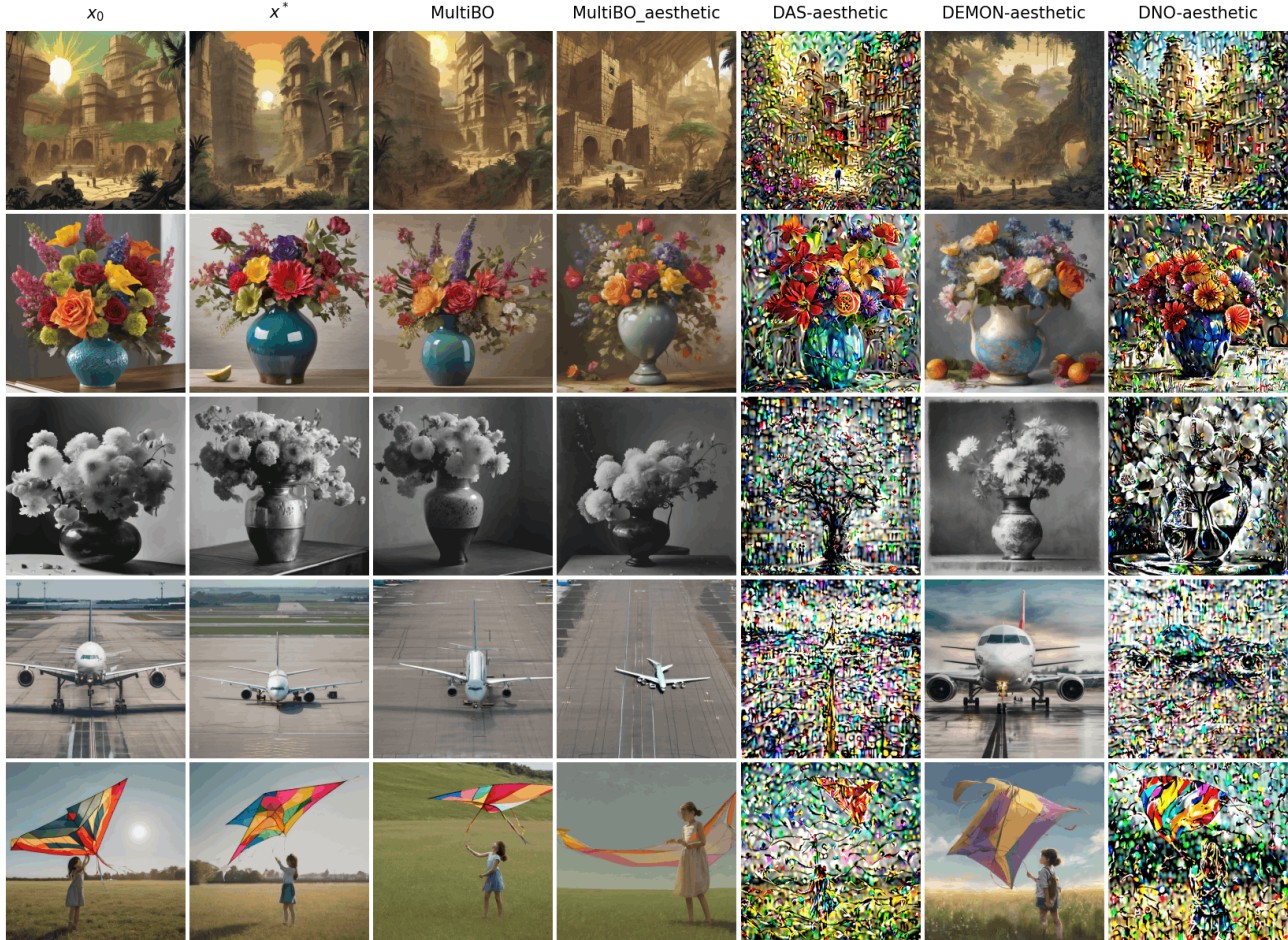

*Figure 20.* Qualitative results comparing **MultiBO** ($B = 50$), **MultiBO**_Aesthetic, DAS_Aesthetic, DEMON_Aesthetic, and DNO_Aesthetic. For prompts: *"A comic book illustration by John Kirby depicting a jungle fortress surrounded by dirt walls in a marketplace setting with cinematic rays of sunlight.", "The smooth, glossy finish of the ceramic vase accentuated the vibrant colors of the flowers, a stunning centerpiece of beauty.", "A black and white photo of a steam of flowers inside a vase.", "An airplane on the runway of an airport.", "A girl is holding a large kite on a grassy field."*

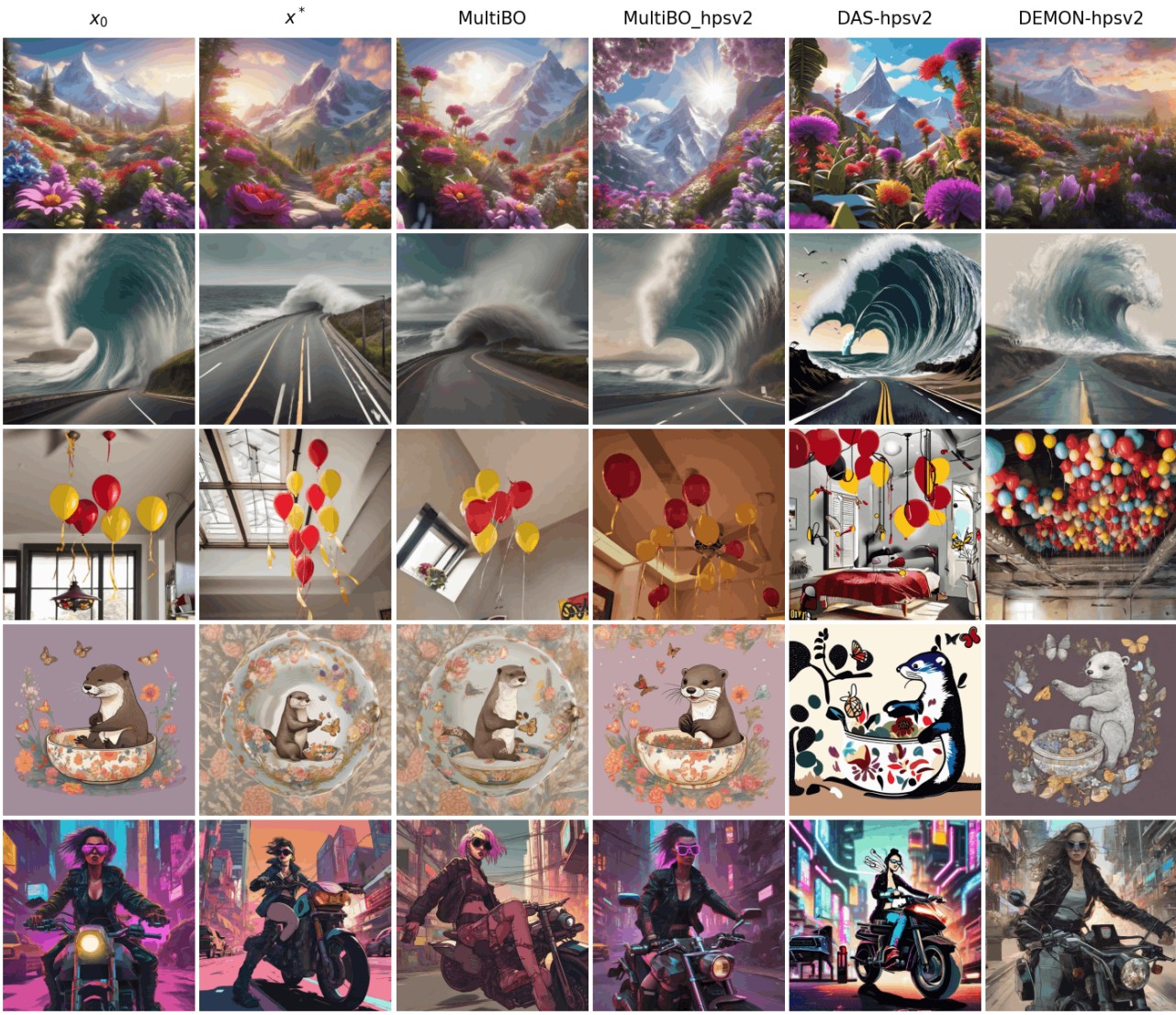

*Figure 21.* Qualitative results comparing **MultiBO** ($B = 50$), **MultiBO**$_{\text{HPSv2}}$, DAS$_{\text{HPSv2}}$, DEMON$_{\text{HPSv2}}$, and DNO$_{\text{HPSv2}}$. For prompts: *"a stunning 3d render of towering, giant blooming plants with vibrant, colorful flowers on a picturesque mountain landscape. sunlight dances on the petals, creating an enchanting scene as the wind gently sways the plants, with snow-capped peaks in the distance.", "a tidal wave approaching a coastal road", "red and yellow balloons hanging from a ceiling fan", "A flower patterned otter is playing with a butterfly shaped bowl", "A cyberpunk woman on a motorbike drives away down a street while wearing sunglasses."*

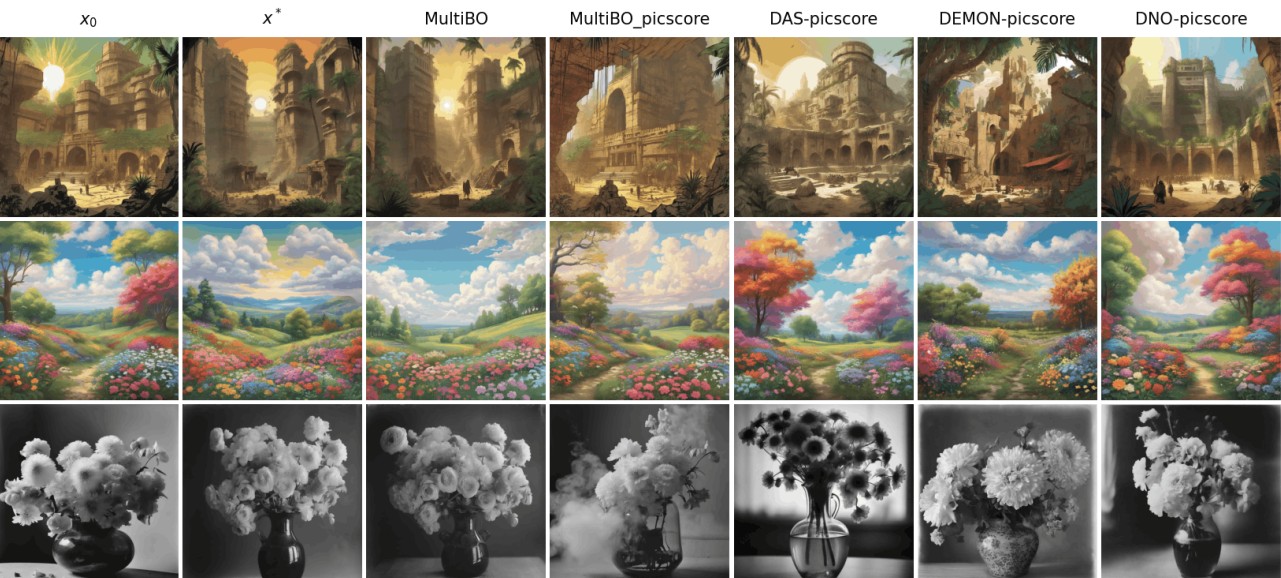

*Figure 22.* Qualitative results comparing **MultiBO** ($B = 50$), **MultiBO**$_{\text{PicScore}}$, DAS$_{\text{PicScore}}$, DEMON$_{\text{PicScore}}$, and DNO$_{\text{PicScore}}$. For prompts: *"A comic book illustration by John Kirby depicting a jungle fortress surrounded by dirt walls in a marketplace setting with cinematic rays of sunlight.", "A peaceful, nature-filled landscape with vibrant flowers and trees and a serene cloud-filled sky.", "A black and white photo of a steam of flowers inside a vase."*

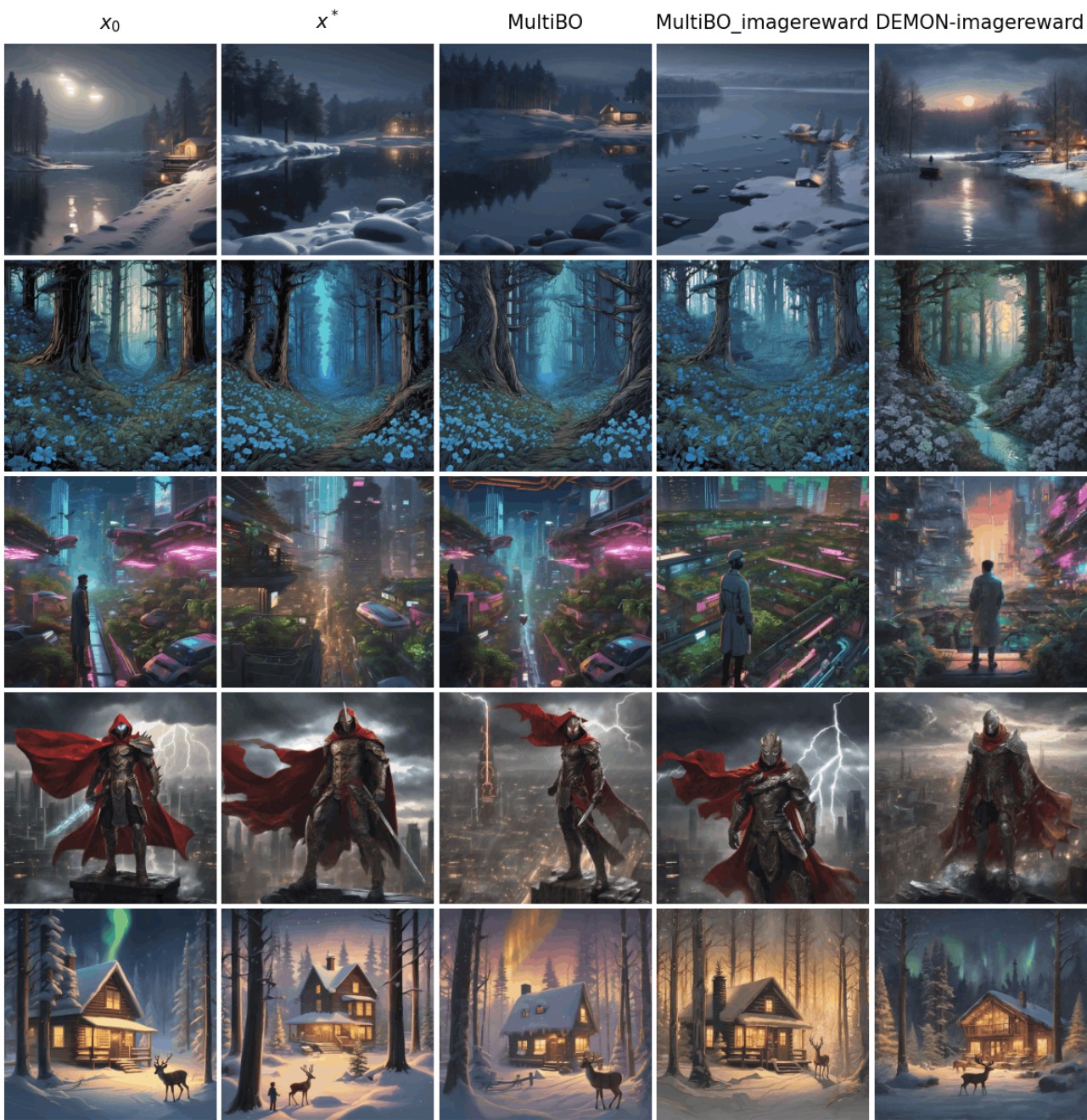

*Figure 23.* Qualitative results comparing **MultiBO** ($B = 50$), **MultiBO**$_{\text{ImageReward}}$, DEMON$_{\text{ImageReward}}$. For prompts: *"A vividly realistic depiction of a snowy Swedish lake at night with hyper-detailed, cinematic-level artistry showcased on ArtStation.", "A forest with blue flowers illustrated in a digital matte style by Dan Mumford and M.W Kaluta.", "On the rooftop of a skyscraper in a bustling cyberpunk city, a figure in a trench coat and neon-lit visor stands amidst a garden of bio-luminescent plants, overlooking the maze of flying cars and towering holograms. Robotic birds flit among the foliage, digital billboards flash advertisements in the distance.", "Amidst a stormy, apocalyptic skyline, a masked warrior stands resolute, adorned in intricate armor and a flowing cape. Lightning illuminates the dark clouds behind him, highlighting his steely determination. With a futuristic city in ruins at his back and a red sword in hand, he embodies the fusion of ancient valor and advanced technology, ready to face the chaos ahead.", "A cozy winter cabin in a snowy forest at night. Warm yellow lights glow from the windows, and smoke gently rises from the chimney. A deer stands near the trees, watching as a child builds a snowman. In the sky, the northern lights shimmer above the treetops."*

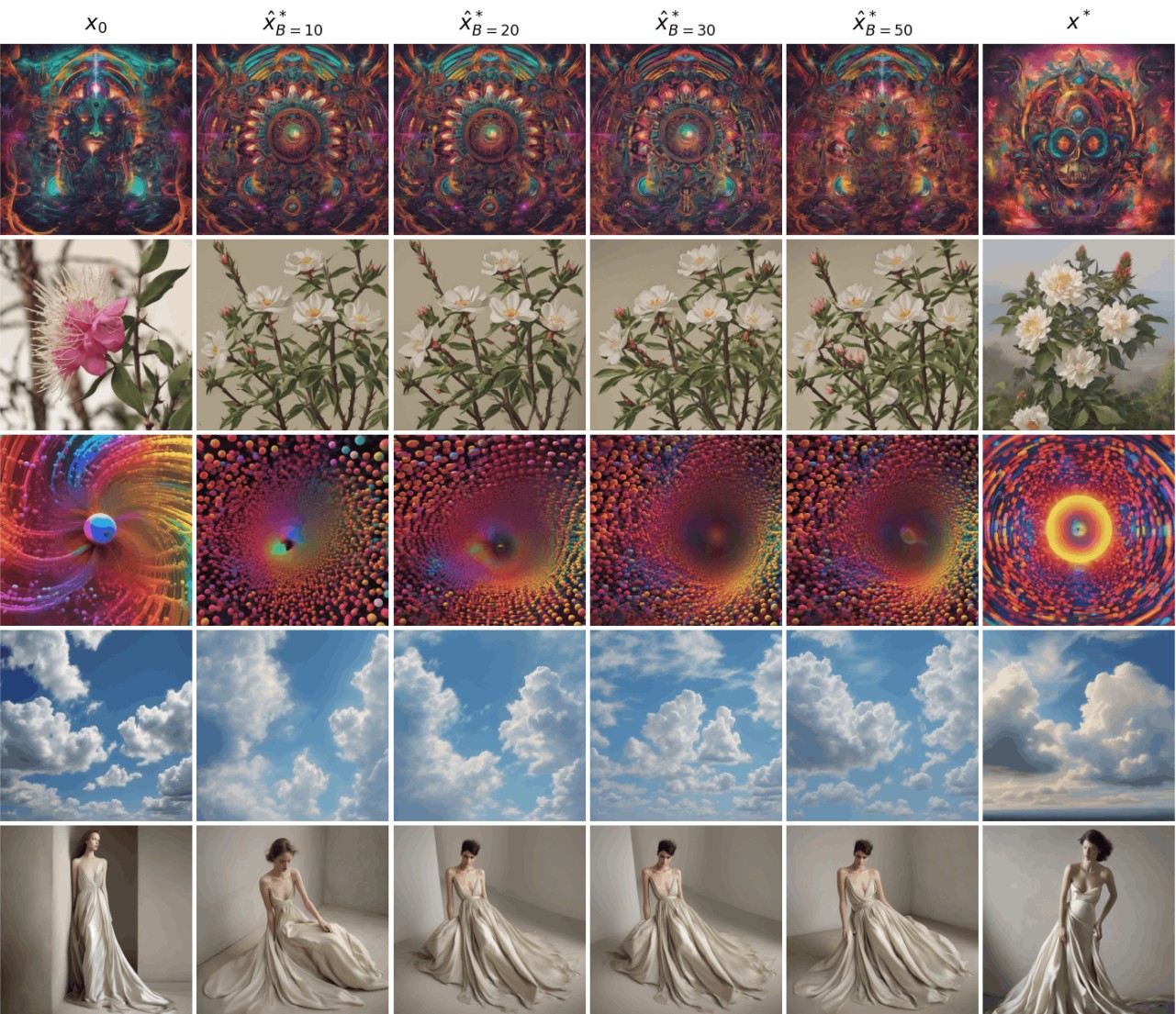

*Figure 24.* Qualitative Examples when **MultiBO** falls short of reaching $x^*$ in $B = 50$ iterations. For prompts: *"Psytrance artwork featuring octane design.", "Beautiful flowering plant with big flowers.", "an electron cloud model is displayed in vibrant colors with a light spectrum background, showcasing the probability distribution of electrons around the nucleus. the image resembles digital art with pixelated elements, bringing a modern, educational twist to atomic structure visualization.", "The soft, fluffy clouds drifted lazily across the blue sky, a canvas of endless possibilities and imagination.", "The smooth silk gown flowed over the delicate skin and the rough floor."*

