# OpenReview forum: "Personalized Image Generation via Human-in-the-loop Bayesian Optimization"
_ICML.cc/2026/Conference — ICML 2026 regular_

### Official Review · Reviewer_ereu · 2026-03-11

**Soundness:** 3
**Presentation:** 3
**Significance:** 3
**Originality:** 3
**Overall Recommendation:** 4
**Confidence:** 4

**Summary:**

The paper introduces MultiBO, a training-free, human-in-the-loop framework for personalized image generation. The core of the method is a Multi-choice Preferential Bayesian Optimization (Multi-choicePBO) that leverages "N-out-of-K" human feedback to iteratively refine the generation process. By optimizing geometric transformations (warping) within the self-attention layers of a diffusion model and utilizing a Dynamic Balanced Subspace (DBS) acquisition function, the system navigates the image manifold to align with user preferences without the need for offline fine-tuning.

**Compliance With Llm Reviewing Policy:**

Affirmed.

**Final Justification:**

I have read the author's rebuttal and the comments from other reviewers. The author's rebuttal resolved my issue, and I have no further questions. I will maintain my positive rating for this paper.

**Key Questions For Authors:**

- What are the limits of "attention warping"? For instance, can it facilitate fundamental color or material changes that are not present in the initial? Furthermore, did the authors observe any instances of "optimization collapse" or artifacting during the warping process?
- Have the authors explored methods to reduce the required number of feedback rounds?
- How do the authors suggest mitigating the sensitivity to the random seed to ensure a robust user experience? Could the random seed itself be treated as a parameter in the optimization space?
- Could the authors provide a more detailed analysis of the computational overhead?

**Limitations:**

yes

**Strengths And Weaknesses:**

# Strength
- The paper is exceptionally well-structured and logical. The transition from the theoretical Bayesian framework to its practical implementation in diffusion models is clear and easy to follow.
- Utilizing test-time optimization based on human preference is a highly practical approach to hyper-personalization, offering a solution for scenarios where language-based control reaches its expressive limits.
- The paper provides extensive experimental results, involving a wide range of baselines and ablation studies. The inclusion of qualitative scores from 30 users adds a layer of objective validation to the performance claims.
# Weakness
- The authors outline a significant issue regarding the high dimensionality of pixel space; however, the chosen alternative—attention warping—raises concerns. The stability and effectiveness of such geometric manipulations in the attention layer are questionable, as they may not be sufficient to handle complex semantic or textural changes.
- The requirement for 50 rounds of human feedback is quite demanding for an interactive system. This high cognitive and temporal burden on the user may limit the method's utility in real-world creative applications.
- Based on the ablation studies in Figure 5, the optimization results seem significantly influenced by the initial random seed. This suggests that the final output may be inconsistent or highly dependent on the starting point of the search.

---

> ### Author Rebuttal · Authors · 2026-03-31
>
> New results in this rebuttal phase: https://annonanom125.github.io/annonAnomrepo/rebuttal.html
>
> We reference them as **R.Table** or **R.Figure**.
>
> **Q1**:
>
> Attention mechanisms are data-driven heuristics that dynamically adjust feature importance, allowing the model to control relevant parts of the generated image. Characterizing the general limitations of attention-warping is difficult. However, we have empirically identified the most commonly occurring failure scenarios, broadly categorized into two groups:
>
> 1. **Conflicting prompt**: If there is mismatch between prompt and target $x^{\ast}$ (e.g.,  if the prompt says “a cat on a table”, but $x^{\ast}$ has the cat below the table), then fixing this mismatch is beyond the capabilities of attention-warping.
> 2. **Model mistakes**: If the prompt is good but the model makes mistakes (e.g., prompt has “a bird” but the starting image $x_0$ is missing a bird), then attention-warping often falters. There are some cases when it can correct these errors but at the expense of higher number of queries to users (i.e., higher B) — see R.Fig. 7.
>
> R.Fig. 7. shows that attention warping can facilitate changes to color, tone, shape, etc. Particularly, in row 2 , a photo realistic image of pink flowers is changed to a painting style with white flowers. In row 3, MultiBO adds color to a BW image.
>
> Warping artifacts (R.Fig. 8) can occur in the initial optimization stages.  However, artifact images are never chosen by the user and such warps are ignored in successive iterations. Even if all choices contain artifacts, the user may pick the “least” bad image, and acquisition-exploration prevents this artifact from propagating through the subsequent iterations.
>
> **Q2**:
>
> We have thought about this problem because we do believe that iterations should reduce before MultiBO becomes ready for the real-world. Several ideas are on our radar for follow-up work:
>
> 1. **Custom generator**: The sample budget may be reduced by using Reward-tuned diffusion models as the base generator.
> 2. **Better Prior**: MultiwiseGPR models from past users (collected from past personalization experiments) as human population models. These models can serve as informed priors to the current user’s MultiwiseGPR model. *The intuition is that rather than starting from scratch, MultiBO starts with a broad understanding of human preference that is then customized to a specific user.*
> 3. **Kernel learning**: The user’s feedback can be utilized to also learn better GPR kernels, which can then serve to learn the black-box function more efficiently, ultimately reducing B.
>
> This paper has focused on the core problem of getting close to $x^{\ast}$ (which itself entails multiple challenges); improving efficiency is our next step.
>
> **Q3**:
>
> MultiBO is sensitive to how well the prompt covers all objects in the desired $x^{\ast}$ and how well the model is able to capture all objects and attributes in the prompt. If these conditions are well satisfied, then MultiBO is not too sensitive to seeds (see R.Fig. 4).
>
> Subsuming the seed-selection process into MultiBO optimization framework sounds interesting, although we would need to carefully handle the very high-dimensional and discontinuous seed-space. This opens a heavy re-design of the whole algorithm, a topic we would like to think more about.
>
> R.Figs. 4, 5, 6 show qualitative results on different randomization scenarios to highlight MultiBO’s robustness to seeds, and the corresponding quantitative metrics are reported in Table 1 in the main paper. These together subsume all randomization scenarios for different prompts from the dataset. These figures highlight the robustness of MultiBO in personalizing to a specific target despite the starting point (within reason) and vice versa.
>
> **Q4**:
>
> MultiBO trades off latency for hyper-personalization, opening a new capability in image generation. R. Tables 1 and 2 report results from memory and timing analysis of the end-to-end MultiBO workflow. Below are salient points,
>
> - MultiBO is lightweight (same memory usage as base SDXL and negligible processing times). It converges in about $B=32$ on average. DEMON reports a lower B because it converges quickly to a suboptimal image (not close to $x^{\ast}$, the worse LPIPS and CLIP-I scores highlight this). Fig. 4 (main paper) also shows this.
> - Moreover, we observe that the user feedback time reduces as B increases. The user becomes more confident in what to look for in choices, and as MultiBO converges, the outliers to reject appear prominent and easily identifiable (see R.Table 2).
> - The average wall clock time is about 20 mins, comparable to DEMON.  Thus, by expending 15-20 mins the user can customize the generation to $x^{\ast}$.
> - MultiBO_LPIPS is much faster because it uses LPIPS metric as a proxy for human choice, taking $x^{\ast}$ as input. Fig. 11(main paper) shows that even with knowledge of $x^{\ast}$, its results fall short due to limitations of LPIPS metric.

---

> > ### Author Rebuttal · Reviewer_ereu · 2026-04-02
> >
> > I have read the author's rebuttal and the comments from other reviewers. The author's rebuttal resolved my issue, and I have no further questions. I will maintain my positive rating for this paper.

---

> > > ### Author Response · Authors · 2026-04-06
> > >
> > > Thank you once again for your thoughtful review. We appreciate the time and effort you have dedicated to reviewing our paper. We are greatly encouraged by your insightful questions. Please let us know if we can provide answers to any remaining questions and potentially improve the scores.

---

### Official Review · Reviewer_iLVr · 2026-03-11

**Soundness:** 3
**Presentation:** 4
**Significance:** 3
**Originality:** 4
**Overall Recommendation:** 4
**Confidence:** 3

**Summary:**

The background of this paper is to optimize the image generation process by using human feedback. The authors proposed a human-in-the-loop Bayesian optimization problem called MultiBO. This method combines multi-choice preference feedback with optimization in a self-attention warping space. The goal is to use a small number of user choices to steer the image toward the target. The paper reports better target alignment and better human evaluation than several baselines.

Overall, I think this is a good paper with impressive presentation and results.

**Compliance With Llm Reviewing Policy:**

Affirmed.

**Key Questions For Authors:**

no questions

**Limitations:**

yes

**Strengths And Weaknesses:**

## Strengths

1. This paper studies a real problem. Prompting alone often does not get users to their desired output.
2. And their main idea makes sense. Asking users to pick the best image is easier than asking for a score.
3. The attention-space optimization is a reasonable design choice. It is more structured than searching in raw pixel or latent space.
4. The paper includes useful ablations on pairwise vs multi-choice feedback, number of choices, layers, and timesteps.
5. The presentation is clear, and it's easy for readers to understand.


## Weakness

1. The task setup may not fully match the claim. The paper is framed around a user’s imagined target image. But the experiments use an actual target image from a dataset. The system is then judged by how close it gets to that visible target. That is not the same thing as helping a user reach a mental image that only exists in their head. The evaluation can be stronger with real human experiments.

2. The human burden is a bit high. The method uses 50 rounds and 4 choices per round. That means a user may look at around 200 images for one run. That would be a lot.

---

> ### Author Rebuttal · Authors · 2026-03-31
>
> New results in this rebuttal phase: https://annonanom125.github.io/annonAnomrepo/rebuttal.html
>
> We reference them as **R.Table** or **R.Figure**.
>
> **W1**:
>
> We had to pick a visible image $x^{\ast}$ as a target only for the purposes of performance evaluation. If we performed a completely real-world experiment, where a user imagines an image and responds to the queries, we would not be able to assess how close MultiBO’s output $\hat{x}^{\ast}$ is to that imagined image $x^{\ast}$.
>
> We understand that in a real-world setting, when the user has $x^{\ast}$ in her imagination, the reliability of preferences may get affected (since the imagined image is not visible, hence the user may not be totally consistent). However, given MultiBO has inherent robustness to noise, we find that MultiBO is able to perform robustly even when 30% of the preferences are chosen at random (see response Q4 to Reviewer **[2aqy]**).
>
> Please let us know if we misunderstood your question.
>
> **W2**:
>
> We agree that the cognitive load is high and we certainly want to minimize it. However, anecdotally, we observe that the user’s cognitive load reduces over time. This is reflected in our empirical results in R.Table 2 where the user takes less time in later iterations to decide on their preferred image.
>
> Our hypothesis is that the user becomes more confident and adept in what to look for when selecting preferred images.  As MultiBO progresses, poor choices — images lacking user’s requirements — easily stand out and the preference choice is easier to identify from the $K$ choices.
>
> In the future, we will work on explicitly modeling the user’s cognitive load and augment the algorithm to account for it. The future work section in the paper outlines some of these possible directions. To the best of our knowledge, this paper is one of the first attempts at hyper-personalization and there is certainly room for improvement.

---

> > ### Author Rebuttal · Reviewer_iLVr · 2026-03-31
> >
> > I don't have other concerns.

---

> > > ### Author Response · Authors · 2026-04-06
> > >
> > > Thank you once again for your thoughtful review. We appreciate the time and effort you have dedicated to reviewing our paper. Please let us know if we can provide answers to any remaining questions and potentially improve the scores.

---

### Official Review · Reviewer_9bEh · 2026-03-13

**Soundness:** 4
**Presentation:** 3
**Significance:** 4
**Originality:** 3
**Overall Recommendation:** 5
**Confidence:** 3

**Summary:**

This paper proposes MultiBO, a Bayesian optimization framework designed to generate images with diffusion models that better match a user’s mental image. The authors point out that prompt-based generation alone often fails to produce images that align with a user's preferences. To address this limitation, the proposed method incorporates human preference feedback into the generation process through Bayesian optimization. However, applying Bayesian optimization during diffusion-based image generation raises two main challenges: (i) the number of user feedback interactions is limited, and (ii) optimization becomes difficult in high-dimensional parameter spaces. To address the first issue, the proposed method uses an N-out-of-K selection scheme, which allows the system to extract more information from a single round of user feedback. To address the second issue, the optimization is restricted to transformations of the self-attention Q, K, and V features, thereby reducing the dimensionality of the optimization problem.

**Compliance With Llm Reviewing Policy:**

Affirmed.

**Final Justification:**

After carefully reviewing the paper, the authors' rebuttal, and the other reviews, I maintain my strong positive assessment and recommend acceptance.

The originality and significance of this work are highly commendable. While previous approaches typically rely on proxy reward models to guide image generation, this paper innovates by directly incorporating human preference feedback into the optimization process. This direct alignment elegantly captures subtle human nuances that approximate evaluation metrics often miss, resulting in generated images that better align with actual user preferences.

The soundness of the proposed methodology is another major strength. Applying Bayesian optimization directly during generation is notoriously difficult due to the limited number of user interactions and the high-dimensional optimization space. The authors smartly overcome these practical bottlenecks by implementing an N-out-of-K selection scheme to maximize the information gained per interaction, and by efficiently restricting the optimization space to self-attention features (Q, K, V). Through these well-thought-out design choices, the paper presents a highly practical Bayesian optimization framework for diffusion models. Furthermore, this framework is backed by extensive experiments against a wide range of baselines, providing strong quantitative evidence that direct human feedback can indeed outperform traditional reward-model-based guidance.

Since I did not have any major concerns during the initial review, my primary focus during the rebuttal phase was observing the broader discussion. The authors' responses to other reviewers were solid and only reinforced my confidence in the robustness of the method. Given its innovative framework, practical engineering, and strong empirical results, I believe this is a highly valuable contribution to ICML.

**Key Questions For Authors:**

I do not have any major questions for the authors. The paper is generally clear, and the experimental results sufficiently support the claims made in the paper.

**Limitations:**

yes

**Strengths And Weaknesses:**

Strengths
- Image generation guided by human feedback: Previous approaches typically optimize reward models to guide generation. In contrast, the proposed method directly incorporates human preference feedback into the optimization process, enabling generation that better aligns with user preferences. The experimental results show that the proposed method achieves better quantitative performance than methods based on reward models. This result is particularly interesting, as it supports the authors’ claim that humans can serve as a more sophisticated guidance signal capable of capturing subtle nuances that approximate evaluation metrics may fail to capture.
- A practical Bayesian optimization framework for use during generation: Applying Bayesian optimization during image generation introduces two key challenges: (i) the limited number of user feedback interactions and (ii) the high dimensionality of the optimization space. The proposed method addresses these issues by (i) increasing the information gained per interaction using an N-out-of-K selection scheme, and (ii) restricting the optimization space to self-attention features (Q, K, V). Through these design choices, the paper presents a practically usable Bayesian optimization framework for diffusion-based image generation.
- Extensive experiments and strong empirical results: The paper evaluates the proposed method against a wide range of baselines and demonstrates its effectiveness through comprehensive experiments. The results provide strong empirical support for the authors’ hypothesis that human feedback can serve as a more effective guidance signal than reward models.

Weakness
- Typos and presentation issues: (1) In Eqs. (6) and (7), the notation $k=1$ appears to be incorrect; it may have been intended to be $i=1$, (2) There is text placed above Figure 4 that appears to overlap with the figure area, which slightly reduces readability.

---

> ### Author Rebuttal · Authors · 2026-03-31
>
> Thank you for catching the typos; we apologize. We will address all of them in the final manuscript.

---

> > ### Author Rebuttal · Reviewer_9bEh · 2026-04-03
> >
> > Thank you for your efforts during the rebuttal phase. I keep my positive recommendation.

---

### Official Review · Reviewer_2aqy · 2026-03-14

**Soundness:** 3
**Presentation:** 3
**Significance:** 3
**Originality:** 3
**Overall Recommendation:** 4
**Confidence:** 4

**Summary:**

This work studies personalized image generation when a user has a target image in mind but cannot fully express it with prompts alone. The method, MultiBO, uses multi-choice human preference feedback inside a Bayesian optimization loop over attention-space edits. Experiments suggest that this setup gets closer to the hidden target than reward-model baselines and a few other human-in-the-loop alternatives, though the user burden is still substantial.

**Compliance With Llm Reviewing Policy:**

Affirmed.

**Key Questions For Authors:**

see weakness and:
- How much wall-clock time do users typically spend to complete `B = 50` rounds in practice?
- Where does performance start to flatten out if the feedback budget is reduced below 50?
- How robust is MultiBO when the initial prompted image is far from the user’s target rather than already somewhat close?
- Can the method handle users whose preferences are inconsistent across rounds, or does the BO loop become brittle quickly?

**Limitations:**

Yes. Section 6 is reasonably candid about weak preference signals near convergence and the restrictions imposed by the attention-domain parameterization.

**Strengths And Weaknesses:**

Pro:
- The task is well motivated, since there is a real gap between prompt following and matching the exact image a user wants.
- The comparison against `MultiBOCLIP-I2I` and `MultiBOLPIPS` is convincing and supports the claim that human feedback is doing more than approximating one fixed metric.
- The paper is candid in Section 6 about failure cases and about the limits of the attention-space parameterization.

Con:
- Fifty rounds of user feedback is still a lot, so the method is expensive in human time even if it performs well.
- Some baselines are not perfectly matched because training-based preference methods solve a different, more generic alignment problem than instance-specific target matching.
- The method seems constrained by its edit space, and Section 6 suggests that this is a real bottleneck rather than a minor detail.
- The paper would be stronger with a clearer runtime discussion for the full human-in-the-loop workflow.

---

> ### Author Rebuttal · Authors · 2026-03-31
>
> New results in this rebuttal phase: https://annonanom125.github.io/annonAnomrepo/rebuttal.html
> We reference them as **R.Table** or **R.Fig**.
>
> **Q1**:
> Users spend between 20-30 minutes to complete B=50 rounds for a target image. Around 65% of images converge in 35 iterations or less (see R.Fig 1(c)), taking around 20 minutes of wall clock time. This is the end-to-end time, including the image generation. Please refer to Table 1 and 2 (in our [rebuttal page](https://annonanom125.github.io/annonAnomrepo/rebuttal.html)) for full time and memory analysis. Additional explanations for these Tables are included in the response Q4 to Reviewer **[ereu]**.
>
> **Q2**:
> On average, the metrics start saturating at B=35, as shown in R.Fig 1(a) and (b). R.Fig 1(c) shows the distribution of number of iterations until convergence for increasing B.
>
> **Q3**:
> A visually far-away starting point obviously makes the problem more challenging. This is particularly true when (1) the prompt misses some objects that should be present in the target image, or (2) when the prompt is good but the model makes mistakes (e.g., an attribute-object mismatch where the generated image has a red apple on a green bowl even though the prompt said “green apple on a red bowl”).
>
> MultiBO is unable to solve case (1) since bringing back missing objects is almost impossible (the space of plausible objects is extremely high and there is no guidance towards correct objects).
>
> However, we find that case (2) can often be corrected, at the expense of higher number of queries to users (i.e., higher B) — see R.Fig. 7.
>
> Finally, when none of the above mistakes occur, and the initial image $x_0$ is simply visually different from $x^{\ast}$ (due to different seeds $x_T)$, then MultiBO is consistently able to get close to $x^{\ast}$. In other words, MultiBO is not sensitive to seeds, so visually different starting images (R.Fig. 4) lead to convergence.
>
> **Q4**:
> Thanks for raising this important question. We report new unreliability experiments in R.Table.3 and R.Fig. 3.
>
> In these experiments, a fraction of user preferences are replaced with random inputs to MultiBO, as if the users are uniform randomly selecting the images. This fraction varies from 10% to 75% of the iterations. We found the performance is robust up to 30%. The robustness is attributed to Multiwise-GPR’s inherent modeling of noise when accepting a user’s preference. Quantitatively (see R.Table.3), MultiBO with inconsistent input is still comparable to other baselines in Table 1 (main paper), indicating that MultiBO learns broad human preference despite unreliable feedback.

---

> > ### Author Rebuttal · Reviewer_2aqy · 2026-04-03
> >
> > I have read the author's rebuttal and the comments from other reviewers. The author's rebuttal resolved my issue, and I have no further questions. I will maintain my positive rating for this paper.

---

> > > ### Author Response · Authors · 2026-04-06
> > >
> > > Thank you once again for your thoughtful review. We appreciate the time and effort you have dedicated to reviewing our paper. We are greatly encouraged by your insightful questions. Please let us know if we can provide answers to any remaining questions and potentially improve the scores.

---

### Decision · Program_Chairs · 2026-04-30

**Decision:**

Accept (regular)

**Comment:**

This paper suggested personalized image generation by human-in-the-loop protocol (multi-choice human preference feedback).

All the Reviewers observed the obvious merits from this work : human feedback is doing well than using one fixed metric, robust Bayesian optimization, significant experiments validations.

Some of weakness are raised as follows : task is not well match to the claim (iLVr), human load (2aqy, iLVr, ereu), missing runtime (2aqy), variation from initial random seed(ereu), instability on geometric manipulation(ereu), typos(9bEh). After rebuttal responses, all the reviewers agreed that their concerns are fully resolved without any partial disagreement.

From this clear acknowledgement from all the reviewers, I recommend this paper's acceptance.